# Curcumin in Inflammatory Complications: Therapeutic Applications and Clinical Evidence

**DOI:** 10.3390/ijms26199366

**Published:** 2025-09-25

**Authors:** Amber Zafar, Divya Lahori, Aleeza F. Namit, Zackery Paxton, Neha Ratna, Dallin Thornton, Kota V. Ramana

**Affiliations:** Department of Biomedical Sciences, Noorda College of Osteopathic Medicine, Provo, UT 84045, USAdo27.drlahori@noordacom.org (D.L.);

**Keywords:** curcumin, turmeric, inflammation, inflammatory complications, treatment, antioxidant

## Abstract

Curcumin is a diarylheptanoid polyphenol compound derived from the plant species *Curcuma longa*. For thousands of years, it has been used as a dietary supplement, food coloring agent, and natural antibiotic in many Asian countries. Recent studies have also investigated its potential therapeutic role in a variety of inflammatory diseases, including osteoarthritis, asthma, chronic obstructive pulmonary disease, atherosclerosis, irritable bowel syndrome, sepsis, atopic dermatitis, and psoriasis. Although individual studies have reported beneficial effects, a comprehensive discussion on findings across these conditions has been lacking. This review systematically evaluates the therapeutic potential of curcumin in inflammatory diseases. Literature was sourced through a PubMed search using relevant terms such as curcumin, treatment, and the names of each targeted disease over the past two decades. We discussed the key findings on how curcumin administration was associated with improvements in disease markers, symptom relief, or progression delay. Despite promising research outcomes, the current evidence underscores the need for more robust, large-scale studies to confirm these effects and guide the clinical applications of curcumin in managing inflammatory disorders.

## 1. Introduction

Inflammation is an initial biological response that serves as a defense mechanism against pathogen attacks on the body, allergens, and injury, leading to tissue damage and dysfunction. Most of the time, acute inflammation, which serves as a protective mechanism, initiates the healing of damaged cells or fights against pathogens. However, chronic inflammation can lead to the increased production of inflammatory cytokines, chemokines, and growth factors, which, through an autocrine and paracrine manner, activate several cellular signaling pathways, thereby contributing to the pathogenesis. Inflammation is the major contributor to various diseases, including bacterial infections, cardiovascular conditions, metabolic disorders, autoimmune syndromes, neurodegenerative diseases, and cancer [1,2,3]. In the past few decades, various anti-inflammatory drugs have been developed and tested to control these complications [4,5]. However, some of the synthetic anti-inflammatory drugs, such as corticosteroids and non-steroidal anti-inflammatory drugs, have several side effects, and prolonged use can cause tissue damage and impair quality of life [6,7]. Therefore, recent studies have focused on isolating and characterizing several plant-based, naturally occurring compounds with antioxidant and anti-inflammatory properties [8,9,10]. Some of these agents have undergone several preclinical and clinical studies [8,9,10]. Furthermore, due to their good safety and bioavailability, some of them are being used as food supplements, potential adjuncts, or alternatives to conventional pharmacological therapies. One of the most extensively investigated compounds is curcumin. Curcumin is a polyphenolic compound derived from the roots of *Curcuma longa* (turmeric). It has garnered significant attention due to its pleiotropic therapeutic properties and favorable safety profile.

Turmeric contains a mixture of curcuminoids such as curcumin, desmethoxycurcumin, and bisdemethoxycurcumin [11]. It has been used as a spice, turmeric, for thousands of years in several Asian countries. Indeed, it has been used as an anti-inflammatory and anti-microbial compound in several traditional medicinal practices [12,13,14]. Due to this, in the past few decades, several investigators have shown a strong research interest in understanding the molecular mechanisms of action of curcumin and its additional beneficial effects in preventing various complications. These numerous in vitro and in vivo studies indicate that, in addition to its anti-inflammatory and anti-microbial properties, curcumin also possesses anti-carcinogenic, anti-proliferative, and antioxidative properties [15,16,17]. Furthermore, several studies have demonstrated that curcumin, as an antioxidant, targets free radicals and, as an anti-inflammatory agent, inhibits the activation of inflammatory proteins [18,19,20]. It has been shown to prevent activation of redox-sensitive transcriptional factors such as NF-κB and AP-1 and suppresses the production of proinflammatory cytokines in both preclinical and clinical studies [18,19,20]. A few studies have also shown that curcumin, either alone or in combination with other drugs, prevents NLRP3-signalosome activation and innate immune responses [21,22,23,24]. These studies suggest that curcumin exhibits pleiotropic properties, influencing multiple molecular pathways simultaneously and demonstrating therapeutic significance in various diseases.

Furthermore, recent clinical studies have also indicated that curcumin could play a potential role in regulating several inflammation-associated complications in the body [25,26,27]. The therapeutic effects of curcumin have been documented in various diseases, including metabolic disorders, neurodegenerative disorders such as Alzheimer’s disease and Parkinson’s disease, asthma, chronic obstructive pulmonary disease, cancer, diabetes, and infectious diseases like sepsis, and various autoimmune and inflammatory diseases [25,26,27].

Although curcumin has been shown to be effective in treating multiple complications, its efficacy in clinical use has faced a few challenges. The poor water solubility and low bioavailability are the primary concerns [28]. To avoid these challenges, several investigators have developed and tested novel curcumin delivery systems [29,30]. Some of those delivery systems include liposomes, nanoparticles, phospholipid complexes, and curcumin conjugates. Several studies have suggested that nanoformulation-based delivery has significant improvements in bioavailability and solubility [30]. Furthermore, such studies facilitated multiple clinical trials to evaluate the therapeutic efficacy of curcumin in various inflammatory conditions. Indeed, several randomized controlled trials (RCTs) have indicated the safety, effectiveness, and tolerability of curcumin as a therapeutic agent [31]. These studies also confirmed the ability of curcumin to prevent multiple pathologies with significant health benefits. We aim to understand the significance of phytochemicals in integrative medicine and the importance of evidence-based approaches in validating natural therapies for inflammatory diseases. In this narrative review article, we discussed the significance of curcumin in various inflammatory complications, specifically in treating six inflammatory diseases such as osteoarthritis (OA), asthma, chronic obstructive pulmonary disease (COPD), atherosclerosis, inflammatory bowel disease (IBD), sepsis, atopic dermatitis (AD), and psoriasis. We believe that by combining traditional knowledge with modern clinical evidence, this article seeks to provide readers with a clear understanding of the therapeutic significance of curcumin in various inflammatory complications. The most relevant articles were selected from PubMed searches for the past decade or so using the keywords curcumin and selective disease. Only empirical studies, peer-reviewed studies, and journal publications were included. Studies were included only if they were published in the last fifteen years, between 2010 and 2025, with a few exceptions from 2002 to 2010. Further, we included only those related to OA, asthma, COPD, atherosclerosis, IBD, sepsis, atopic dermatitis, and psoriasis. We did not include articles from magazines, commentaries, opinion pieces, newsletters, or any research on other complications. Specifically, this article is a narrative review rather than a systematic review because, although we synthesized evidence from multiple peer-reviewed studies on curcumin, this article does not follow a standard protocol for study identification, selection, or appraisal. Furthermore, the literature search was limited to PubMed using selected keywords and restricted to specific years and inflammatory diseases, without a structured process such as PRISMA flow diagrams, risk-of-bias assessments, or meta-analysis. Through this article, we intend to provide a descriptive overview and thematic discussion on curcumin’s use in some important inflammatory complications.

## 2. Curcumin in Treating Osteoarthritis

Joint inflammation, also known as arthritis, is a primary cause of musculoskeletal disability worldwide. It represents multiple subclassifications, and Osteoarthritis (OA) is one subclass that commonly affects elderly individuals. It induces pain with routine motion, resulting in a diminished quality of life. The common factors associated with its onset are either nonmodifiable, meaning the patient has a genetic or congenital predisposition for the disease, or modifiable, meaning peripheral factors are exacerbating the patient’s condition [32]. Under these classifications, aging and obesity were found to be most influential in the emergence of osteoarthritis. As individuals age and gain weight, the activity of proinflammatory markers increases, as well as the pressure on their joints, leading to inflammation [32,33]. There is no complete cure for osteoarthritis, but common therapeutics include exercise, orthopedic aids, pharmacological substances, and surgery. All of which are administered to relieve pain, delay disease progression, and improve the quality of life among patients [34]. Non-steroidal Anti-inflammatory Drugs (NSAIDs) such as meloxicam, naproxen, and diclofenac are commonly prescribed for pain relief, but after long-term use, are known to cause gastrointestinal (GI) complications [35]. This is especially the case in elderly patients with chronic NSAID usage [35,36].

Additionally, surgeries such as knee arthroplasty are performed after the end-stage progression requirements have been met. However, postoperative recovery is hindered by surgery-acquired trauma and noticeable declines in strength and activity that are worsened with age. Nevertheless, current research suggests that preoperative high-intensity strength and balance training may improve patient satisfaction and joint function post-arthroplasty [37].

Consequently, arising from the complexity of osteoarthritis and the long-term effects current treatments have on patients is the need for improved therapeutics in combating the inflammatory and degenerative properties. Curcuminoids are naturally sourced plant derivatives presently under investigation as potential alternatives to the current NSAIDs regimens for patients diagnosed with OA. Shep et al. [38] discussed this alternative form of treatment by investigating the standard NSAID diclofenac compared with curcumin in the treatment of knee OA. Participants aged 38–65 years were assessed for a minimum of 3 months based on their reported pain, as measured by a visual analog scale (VAS), knee injury, and osteoarthritis outcome score (KOOS) subscale [38]. Baseline pain scores, as measured by VAS ratings, for curcumin-treated (500 mg × 3 times/day) patients were 7.84 ± 0.73 and 7.81 ± 0.73 for the diclofenac trial group. Subsequent scoring after 28 days indicated net decreases of 5.93 ± 0.99 and 5.61 ± 0.88 in VAS ratings for the curcumin and diclofenac patient groups, respectively. Moreover, the significant findings in this study were the *p*-values for intra-group comparisons of baseline patient scores and the net change after 28 days. These were *p* = 0.79 and *p* = 0.82, indicating no notable difference in treatment when patients were managed with diclofenac or curcumin. Furthermore, comparable improvements in patients receiving diclofenac versus curcumin treatments were observed, based on KOOS daily living and quality of life scores. These, along with the identical VAS items, strengthen the justification for utilizing curcumin as an alternative form of treatment for OA patients when compared to NSAIDs [38].

As previously discussed, curcumin has garnered attention as a prospective, bioavailable adjuvant treatment for osteoarthritis due to its anti-inflammatory, antioxidant, and non-toxic properties [33,34,35] (Figure 1). Moreover, Mathy-Hartert et al. [39] have investigated the influence of curcumin on specific biomarker expression in articular chondrocytes. In their activated state, articular cells produce interleukin (IL)-1beta, activates inflammatory compounds, and inhibits the production of type II collagen and proteoglycans [39]. The results of the study mentioned above demonstrated that the presence of IL-1β increased the activity of proinflammatory molecules, such as nitric oxide (NO), prostaglandin (PG) E2, IL-6, and IL-8. However, following the addition of increasing curcumin concentrations (0 to 20 μM), both the basal and IL-1β-dependent expression of NO and PGE2 were found to decrease in a dose-dependent manner (*p* < 0.001) after 12-day culture analysis. Indicating a reduced proinflammatory compound production. Additional data presented by Mathy-Hartert et al. [39] identified a similar 12-day culture decline in matrix metalloproteinase (MMP)-3 synthesis (*p* < 0.001). The drop was identical to the one observed from the chondrocyte activation of NO and PGE2; however, there was no significant impact of tissue inhibitor of matrix metalloproteinase (TIMP)-1. This effectively led to the diminished MMP-3 to TIMP-1 ratios (*p* < 0.001), indicative of a decrease in MMP-3 breakdown of the extracellular matrix [39].

Accordingly, in a randomized, double-blind, placebo-controlled, parallel-group trial performed by Panahi et al. [40], they have measured the potential clinical efficacy of curcuminoids in treating OA. The study included 40 volunteers, older than 80 with OA, who were assessed based on changes in their Western Ontario and McMaster Universities Osteoarthritis Index (WOMAC), a VAS, and Lequesne’s pain functional index (LPFI) to quantify each patient’s pain, stiffness, and physical functions for 6 weeks. From the organized data, a significant WOMAC score reduction (*p* < 0.001) was noted within the curcuminoid-treated group for self-reported pain, physical function, and global scoring items. Additionally, there was a significant difference in the reduced global scores between the two groups (*p* < 0.001). The placebo group had a score of 44.6 ± 17.3 before treatment and 40.6 ± 12.6 after treatment, with the curcuminoid-treated group reporting pre- and posttreatment scores of 42.4 ± 18.3 and 25 ± 13, respectively. Similarly, patients who received curcuminoid treatments reported significant overall decreases (*p* < 0.001) in both LPFI and VAS scores by the end of the trials, with no significant differences in LPFI and VAS within the placebo group [40]. In this study, the authors have also measured volunteers’ use of naproxen throughout the trials. Consequently, an 84% decrease in naproxen usage was observed among subjects administered treatment that included curcumin, while only 19% of patients in the placebo-control group reported any significant reduction in NSAID usage [40]. As indicated by these results, curcumin treatments could help reduce OA inflammation by inhibiting NF-κB and inflammatory processes, leading to further pain reduction and improved physical activity.

Furthermore, reductions in NSAIDs in the maintenance of OA were likewise noted in a study that investigated the co-administration of N-palmitoyl-D-glucosamine (PGA)-curcumin in dogs. Furthermore, Dell Rocca et al. [41] observed that when dogs with moderate to severe OA were administered the PGA-curcumin supplement, there was a decrease in meloxicam dosage in 90% of subjects, with up to a 25% reduction compared to previously recorded levels, and no increase in pain according to owners [41]. Correspondingly, 75% of owners reported that their pets’ pain was well-managed with the supplementation above up to 10 weeks after meloxicam discontinuation.

Conversely, impediments affecting the availability and efficiency of curcumin in OA treatment have been documented. One frequent obstacle in the use of curcumin is its bioavailability. When administered orally, therapeutic absorption is diminished, and so is its efficacy. Therefore, current dosage recommendations vary between 1000 and 2000 mg/day [38]. The reason supporting the low bioavailability of curcumin to manage OA can be attributed to known vessel deficiencies within articular cartilage [42]. However, alternative methods for administering curcumin and enhancing its effectiveness have been observed primarily through its co-administration with other compounds. Several studies have noted progress in curcumin potency when formulated with substances such as Piper extract Bioperine, turmeric essential oils, black pepper, and specific metal–organic frameworks (MOFs). Results from these treatments have demonstrated increased curcumin absorption through the GI tract with decreased metabolic elimination [40,41,42].

Meanwhile, a recent randomized controlled trial carried out by Hsueh et al. [43] has found that curcumin decreases serum levels of C-reactive protein and inflammatory cytokines such as TNF-α in patients with knee OA. Another clinical study by Noori et al. [44] indicates that curcumin significantly increases the gene expression of FOXO1 and HAS-MIR-873, suggesting that in knee OA patients, curcumin regulates the Th-17 function. Further, Baharizade et al. [45] have shown that self-nano-emulsifying (SNE) physically cross-linked polyethylene glycol (PEG) organogel (SNE-POG)-mediated delivery of curcumin is effective for knee OA treatment. Topical formulation of curcumin has been shown to have a good analgesic effect on knee OA patients [46]. Few other studies have also suggested the use of combinational therapy, along with curcumin, in preventing OA in patients [47,48] (Table 1). Recently, several other clinical studies have also indicated that curcumin may prove to be an effective tool and therapeutic for treating osteoarthritis, based on its high tolerance and anti-inflammatory properties [49,50,51].

## 3. Curcumin in Treating Asthma and COPD

Bronchial asthma is one of the most widespread, non-communicable respiratory diseases, which impacts over 300 million individuals globally [52]. It is a chronic condition that causes airways to narrow and produce extra mucus, leading to wheezing, coughing, and shortness of breath. Further, Chronic Obstructive Pulmonary Disease (COPD) is the third deadliest disease in the world [53]. COPD constitutes a group of lung diseases that include emphysema and bronchitis. These conditions can lead to the destruction and enlargement of alveoli, causing increased mucus production and inflammation of the airways, respectively [54]. Ultimately, COPD presents similar symptoms to asthma. Asthma is caused by a wide variety of risk factors, including hereditary predispositions, obesity, diet, and environmental factors such as allergens, infections, tobacco smoke, and air pollution [54]. The primary goal of asthma treatment is to suppress inflammation. Asthma can be treated with inhaled corticosteroids (ICS), which provide instant relief, short-acting beta-agonists (SABAs), which rapidly reduce airway bronchoconstriction, and long-acting beta-2-agonists (LABAs) for the maintenance of the airways [55]. These treatments are often combined for the most effective therapy. Bronchodilators and leukotriene modifiers (LTMs) also improve lung function by mitigating the inflammatory side effects of the disease [54,55]. Unfortunately, ICS and LTMs can lead to respiratory tract infections, viral gastroenteritis, ear, nose, and throat maladies, and dysphonia [56,57]. SABAs and LABAs are implicated in a variety of side effects such as tremors, insomnia, nausea, fever, bronchospasm, and dyspepsia, amongst others [57].

On the other hand, smoking and air pollution from wood burning are the most encountered risk factors for COPD [58]. Significant comorbidities that contribute to the severity of COPD are body weight loss, skeletal muscle wasting, cachexia, osteoporosis, heart failure, cardiac ischemia and arrhythmias, anemia, hypoalbuminemia, diabetes, cognitive deficits, and depression [59].

As COPD is an irreversible condition, it is generally treated with bronchodilators and corticosteroids [60,61]. However, LABAs, ICS, and long-acting muscarinic antagonists (LAMAs) also play a role individually and in concert with each other but may produce deleterious cardiovascular effects with long-term use [62,63]. Additionally, muscarinic antagonist beta-2 agonists (MABAs) have also been shown to induce broncho-relaxant effects, but at higher doses, they can cause adverse side effects [63]. PDE4 inhibitors such as Roflumilast have been approved as an add-on therapy for the management of COPD; however, they can lead to clinically significant side effects such as unexplained weight loss, gastrointestinal tract-related conditions, and unwanted psychiatric effects [64]. Thus, this class of drugs is not being further researched or developed.

Exacerbation of these respiratory illnesses often causes morbidity and eventually mortality. Current treatment regimens produce multiple adverse side effects that can affect patients lifelong. Thus, adding a virtually risk-free agent to current regimens may prove more effective in treating both Asthma and COPD with fewer complications. Curcumin is a potent antioxidant and anti-inflammatory compound derived from turmeric that is commonly used in Asian foods [65,66]. Multiple studies have established that the mechanism of action involves targeting free radicals, inhibiting the activation of inflammatory proteins such as NF-κB and AP-1, and suppressing the production of proinflammatory cytokines in both animal and human studies [19,20,21]. In fact, curcumin was also found to elevate the expression levels of aquaporin (AQP) 1 and 5, which further led to a decrease in pulmonary edema and a reduction in symptoms [67,68].

In 2012, Ng et al. [65] conducted a study among 2478 Chinese adults aged 55 years and above and found that curcumin intake at least once a month was significantly associated with better forced expiratory volume in one second (FEV1) and FEV1/FVC ratio, which is a measure of lung function. Furthermore, the effects were more pronounced in current and past smokers as the FEV1 was 9.2–10.3% higher. These results suggested that curcumin could play a protective role against pulmonary damage in those who smoke. Two years later, Abidi et al. [69] conducted a study in India to research the effect of curcumin capsules as an add-on therapy in asthmatic patients. They demonstrated that after thirty days, there was a significant reduction in FEV1 values. Kohli et al. [66] found that an ideal curcumin dose of 200 mg per kg of body weight prevents the allergic airway inflammation commonly associated with Asthma and inhibits the actions of the inflammatory protein NF-κB. Similarly, Aggarwal et al. [70] have also found that curcumin inhibited airway inflammation and cell infiltration in asthma by inhibiting NF-κB, a central mediator of the innate inflammatory response. In a mouse model asthma study, Yang et al. [71] concluded that curcumin activated the Wnt/β-catenin signaling pathway in dendritic cells and asthmatic mouse lungs in a dose-dependent manner, thereby easing asthma symptoms and reducing the associated inflammatory response.

Furthermore, a few studies have also specifically investigated the effectiveness of curcumin on COPD (Figure 2). However, these studies have established a similar effect of decreasing airway inflammation and alleviating comorbidities in patients with COPD. As the primary cause of COPD, smoking was heavily investigated. Regular smoking attracts neutrophils and lymphocytes into the airways, which release inflammatory cytokines and limit airway expansion in patients with COPD [72]. The therapeutic effect of curcumin in reducing inflammation via the inhibition of NF-κB was confirmed by Belvisi et al. [73], who found that this was achieved by modulating the PPARγ/NF-κB signaling pathway. Further, peroxisome proliferator-activated receptors (PPARs) exhibit anti-inflammatory and immunomodulatory properties. They aid in the regulation of proinflammatory gene expression and inflammatory cell functions, which can help alleviate symptoms of Asthma and COPD [74,75]. Lastly, curcumin was found to increase the production of and enhance the action of interleukin, a potent anti-inflammatory cytokine that is generated by both adaptive and innate immune cells [76,77].

In terms of dosage, it seems that while a low dose of curcumin has a protective effect on cell viability in smokers, a high dose can produce a reverse effect on cell viability [78]. Since curcumin is often poorly absorbed and has low bioavailability, this can limit its effectiveness. A few studies have also shown that combining curcumin with piperine, a compound found in black pepper, can increase its bioavailability by 2000%. Moreover, this combination has been revealed to be effective in reducing respiratory ailments associated with COVID-19 [79,80,81]. Similarly, Panahi et al. [82] reported in a randomized clinical study that a combination of curcuminoids and piperine improves oxidative stress markers and COPD assessment test indices in patients with chronic pulmonary complications resulting from sulfur mustard exposure (Table 1).

Furthermore, a randomized, double-blind, controlled study by Manarin et al. [83] has demonstrated that powdered roots of Curcuma longa show decreased use of short-acting β-adrenergic agonists and better asthma control in children and adolescents 3 and 6 months after supplementation. Further, a recent clinical study by Zare’i et al. [84] also suggested that nanocurcumin improves the pulmonary function in COPD patients by decreasing IL-6 and increasing the forced expiration (FEV1) and forced vital capacity (FVC) ratio. Thus, curcumin is an effective supplemental treatment in asthmatic and COPD patients. Asthma and COPD are both obstructive pulmonary diseases that affect millions globally. However, the use of curcumin has been shown not only to downregulate proinflammatory compounds but also to upregulate signaling pathways associated with reducing inflammatory responses in both diseases (Figure 2). While the curcumin dosage may still require further research, its effect in alleviating airway constriction and increasing FEV1 and FEV1/FVC values has been widely accepted as significant.

## 4. Curcumin in Treating Atherosclerosis

Atherosclerosis is a chronic inflammatory disease due to the buildup of plaque in the arteries [85,86,87,88,89]. These plaques form by adhering to the tunica intima and eliciting an inflammatory response that leads to increased plaque buildup [88]. The probability of contracting atherosclerosis is increased when exposed to the following risk factors: hypercholesterolemia (LDL cholesterol), hypertension, diabetes mellitus, smoking, age, and gender.

A recent study suggests that women, especially those who were middle-aged and older, had a higher total plaque burden than men. This study has shown that men consistently had a higher proportion of vulnerable fibrofatty and necrotic core plaques, which are linked to greater cardiac risk. These results suggest critical sex-specific differences in plaque types that could influence carotid artery disease risk assessment and management [90]. Further, smoking has been shown to be a strong risk factor for the development of atherosclerosis and other cardiovascular complications. A recent study by Yao et al. [91] analyzed data from 182,000 participants and found a correlation between smoking intensity and an increase in inflammatory and atherosclerosis biomarkers. Similarly, increased cholesterol levels, hypertension, and type-2 diabetes are also associated with the development of atherosclerosis [92,93,94].

The primary goal of treatment for patients with coronary artery disease (CAD) is to lower blood pressure and decrease the amount of “bad” cholesterol (LDL cholesterol) in the blood [95]. Currently, there are three general treatment options: lifestyle modifications, medications, and surgery. In modifying their lifestyle, patients are counseled to change their diet, exercise, and manage their stress and sleep better. At times, this has proven sufficient, but most often, it requires the additional use of medication. Most medications given to treat atherosclerosis, such as ACE inhibitors and Aspirin, aim to lower blood pressure. Additional drugs, such as statins, are also administered to help reduce the amount of LDL cholesterol in the blood [95,96,97]. These treatments do carry some risks, such as excessive bleeding, muscle pain, memory fuzziness, hyperkalemia, and reduced kidney function.

Cell culture studies investigating the effects of curcumin on atherosclerosis offer valuable insights into potential therapeutic strategies for combating cardiovascular disease [98]. Exposure to curcumin has been shown to regulate several key pathways implicated in atherosclerosis development, including inflammation, oxidative stress, and lipid metabolism [98,99,100] (Figure 3). In another study, macrophages were polarized into M1 by the introduction of LPS and IFN-γ. These cells were then treated with different concentrations of curcumin and assessed by measuring the levels of inflammatory cytokines (TNF-α, IL-6, and IL-12) and evaluating the TLR4 transduction pathway. Conclusions were drawn that curcumin was able to block signal transduction at the receptor level, as well as inhibit the phosphorylation of the MAPK cascade. It was also determined that curcumin, dose-dependently, inhibited M1 macrophage polarization and production of the above-mentioned inflammatory cytokines [101,102]. Curcumin has demonstrated effectiveness in various intracellular processes, mitigating the inflammatory response within macrophages. The primary therapeutic effect of curcumin is the promotion of macrophage polarization to the M2 phenotype by use of IL-4 and IL-13, which regulate the role of macrophages in inflammatory signaling [103].

Additionally, curcumin has been shown to inhibit signal transduction within macrophages, decreasing the inflammatory response (Figure 3). One of the ways this is accomplished is by curcumin’s ability to block homodimerization of the TLR-4 receptor on macrophages as well as its ability to inhibit MAPK in the signal transduction cascade [104]. Finally, there is convincing evidence that shows the ability of curcumin to inhibit foam cell formation, which in turn inhibits CD36, a known biomarker of atherosclerosis [105]. These proposed mechanisms all work together to demonstrate the versatile and elastic properties of curcumin in combating atherosclerosis.

Utilizing various preclinical animal models of atherosclerosis, a few studies have investigated the effects of curcumin supplementation on cardiovascular disease progression [106,107,108]. For example, curcumin has been shown to protect against atherosclerotic lesions induced by a high-fat diet in mice. Similarly, in ApoE mice, curcumin has been shown to inhibit TLR4 expression and macrophage infiltration [108]. Additionally, curcumin also decreased the inflammatory cytokines IL-1β and TNF-α [108]. These studies thus demonstrate that curcumin supplementation can prevent atherosclerosis by reducing plaque formation, specifically by decreasing inflammation and improving lipid profiles [109,110]. Additionally, preclinical animal studies also offered insights into the pharmacokinetics and safety profile of curcumin.

Due to the strong antioxidant and anti-inflammatory properties of curcumin, in vitro and in vivo studies have demonstrated its potential as an effective therapeutic agent in the treatment of atherosclerosis. Few human trials have been conducted with curcumin, and additional studies are needed using focused human trials with large population groups.

A recent randomized controlled trial by Yaikwawong et al. [111] evaluated the effects of curcumin on cardiovascular risk in 227 obese patients with type 2 diabetes over 12 months. They found that curcumin significantly reduced arterial stiffness and improved lipid profiles. Furthermore, this study indicates that curcumin also reduces inflammatory markers, suggesting a beneficial role in lowering the atherogenic risk in this population.

Another randomized, double-blind, placebo-controlled trial by Dastani et al. [112] investigated the effects of nano-curcumin on cardiovascular risk factors in 64 type 2 diabetic patients with mild to moderate coronary artery disease (CAD). They have shown that after 90 days of treatment, nano-curcumin significantly reduced inflammatory markers such as (hs-CRP) and lipoprotein A [LipoPr(a)] levels when compared to placebo. This study suggests that nano-curcumin may help to prevent atherosclerosis progression and reduce the risk of future cardiovascular events in diabetic patients. Similarly, Funamoto et al. [113] have evaluated the effects of Theracurmin^®^ in patients with mild COPD over 24 weeks. They have shown that while most metabolic and cardiovascular markers showed no difference between the treatment and placebo groups, Theracurmin^®^ significantly reduced levels of atherosclerotic AT-LDL (Table 1). Through these preclinical and clinical studies, several attempts were made to understand the molecular mechanisms by which curcumin may attenuate atherosclerotic processes, potentially paving new pathways for the development of novel therapeutic interventions targeting cardiovascular health.

## 5. Curcumin in Treating IBD

Inflammatory Bowel Disease (IBD) is a group of diseases characterized by chronic inflammation of the digestive tract. Ulcerative Colitis (UC) and Crohn’s Disease (CD) are types of IBD. Ulcerative Colitis involves inflammation and ulcers along the colon and rectum. In contrast, Crohn’s Disease is characterized by inflammation of the lining of the digestive tract, primarily in, but not limited to, the small intestine [114]. Both conditions cause a host of symptoms such as diarrhea, rectal bleeding, and abdominal pain. While the exact cause of IBD remains unknown, diet and stress exacerbate this condition, and immune system malfunction, gene mutations, and heredity may play a role. Risk factors include age, race, family history, smoking history, and NSAID use. Smoking can lead to Crohn’s disease but prevents Ulcerative Colitis, and NSAID use can increase the risk of developing IBD [115,116]. The most common patient demographic to be diagnosed with IBD is Caucasians below the age of thirty. This chronic condition affects over 1.2 million adults in the United States. It can lead to complications such as colon cancer, inflammation, blood clots, severe dehydration, fistulas, anal fissures, toxic megacolon, and perforated colon [117,118]. Current treatment options include preventing inflammation to manage symptoms using anti-inflammatory drugs, immunosuppressants, antibiotics, analgesics, and other natural food supplements. Additionally, proctocolectomy or bowel resection with colostomy may be indicated in severe cases. Given the prevalence of IBD and that current treatment regimens, such as medication and surgery, can have side effects, limited efficacy, or be invasive, alternative therapeutic options such as curcumin are being researched as potential treatments.

There are various mechanisms by which curcumin curbs IBD-associated inflammation [119,120,121]. Cell culture studies have demonstrated that curcumin modulates inflammatory pathways and protects against oxidative stress. IBD patients suffer from chronic inflammation and production of proinflammatory factors, resulting in a cycle of intestinal barrier damage and continued inflammation. A study by Wang et al. [122] has demonstrated that curcumin inhibits the activation of inflammatory pathways by blocking the NF-κB and MAPK signaling pathways, resulting in decreased cytokine production. In this cell culture (HT-29) experiment, curcumin induced apoptosis and prevented the growth of human colon cancer cells, HT-29, via a mitochondria-mediated pathway. Similarly, another study by Wei et al. [123] indicated that curcumin prevented DSS-induced colitis in mice by regulating memory B cells and activating the Bcl-6-Syk-BLNK signaling pathway. In another study by Huang et al. [124], curcumin has also been shown to prevent DSS-induced colitis in mice by regulating B-regulatory cells and inhibiting the TLR/Myd88 pathway.

Furthermore, a study by Song et al. [125] has also suggested that curcumin, by reducing the levels of memory helper T cells and inhibiting the IL-7/IL-17R signaling pathway, prevents DSS-induced colitis. Similarly, Wang et al. [126] have also demonstrated that curcumin, by inhibiting T follicular helper cell differentiation, prevents the DSS-induced colitis in mice. Zhao et al. [127] have also indicated that curcumin prevents the activation of dendritic cells by regulating the JAK/STAT/SOCs signaling pathway in colitis induced by 2,4,6-trinitrobenzene sulfonic acid. Furthermore, some studies also suggest that curcumin prevents NF-κB-mediated proinflammatory and NLRP3-mediated innate immune response pathways. For example, Hu et al. [128] indicated that curcumin analog C66 prevents DSS-induced colitis by inhibiting the activation of JNK/NF-κB pathways. On the other hand, Gong et al. [129] have shown that curcumin inhibits NLRP3 inflammasome activation and IL-1β generation in a DSS-induced colitis mouse model. Thus, several preclinical animal studies have suggested that curcumin regulates immune cell modulation and inflammatory response and prevents experimental IBD [130,131].

Recent clinical trials have also explored the effect of curcumin on IBD, measuring factors such as quality of life and inflammation [132,133,134]. Few clinical studies have investigated the use of curcumin as a supplement for the treatment of IBD. In a randomized controlled trial, improvements in Crohn’s disease were observed when patients were treated with curcumin, resulting in decreased inflammatory markers [135]. Both C-Reductive peptide (CRP) and Interleukin-6 (IL-6) levels decreased significantly when a combination of curcumin with mesalamine was used, inducing remission in patients with ulcerative colitis [136]. In addition, they have also shown that 53.8% of patients who supplemented their treatment with curcumin reached remission by week four, and 30% of patients achieved endoscopic remission, compared to 0% in the placebo group. When a compound containing curcuminoids, known as Xanthofen or IQP-CL-101, was used to treat IBS symptoms, patients experienced improvements in IBS-Symptom Severity Score and quality of life after four to eight weeks [137]. It was reported that the treatment group experienced an 8.6% greater reduction in IBS severity than the placebo group at week four and a 32.2% greater improvement in symptoms on the IBS-Global Improvement Scale [137]. Statistically significant results were also seen in quality of life after eight weeks and abdominal pain and discomfort after five weeks [137].

Ulcerative Colitis causes chronic inflammation in the colon, leading to the excessive production of proinflammatory markers and intestinal barrier damage, which further releases inflammatory markers. When curcumin was administered with drug therapy in patients with Ulcerative Colitis, patients reported an improvement in symptoms, as indicated by the Simple Clinical Colitis Activity Index Score, serum CRP concentration, and erythrocyte sedimentation rate (ESR) values, compared to the placebo group [138].

While curcumin shows promising results in controlling IBD based on recent research (Figure 4), further studies are needed to develop the results, conduct risk-benefit assessments, and address challenges such as bioavailability. Curcumin, along with piperine supplementation, has been explored as an avenue for improving the symptoms associated with IBD. An interesting study by Da Paz Martins et al. [139] has reported that supplementation with curcumin and piperine showed significant improvements, including reduced muscle depletion, compared to placebo. These results suggest that curcumin, combined with piperine, may be a beneficial adjuvant therapy for improving muscle health in IBD patients. The use of a derivative of curcumin with high bioavailability and greater absorption, known as Theracurmin, showed promising results in patients with Crohn’s disease by inhibiting NF-κB and thereby decreasing inflammatory cytokines [140]. In this trial, Sugimoto et al. [140] demonstrated that endoscopic reductions in disease, reduced disease activity, and increased healing of anal lesions were observed over 12 weeks in the treatment group. Curcumin has shown minimal side effects; however, studies assessing the long-term safety of curcumin supplementation are necessary to establish a comprehensive safety profile. Curcumin has the potential to be widely accepted as a form of adjuvant therapy in the treatment of IBD. In patients with Crohn’s disease and Ulcerative Colitis (UC), it has been shown to have a range of beneficial effects, including the inhibition of proinflammatory markers and reactive oxygen species, as well as improved clinical symptoms (Table 1). A multicenter, double-blind study by Banerjee et al. [141] evaluated the effect of curcumin plus mesalamine in patients with mild-to-moderate ulcerative colitis (UC). They have shown that patients in the curcumin group achieved clinical remission, whereas none in the placebo group did. The results demonstrate that curcumin, when added to standard treatment, effectively induces remission in UC without added adverse effects. Similarly, Ben-Horin et al. [142] have also shown that the combination of curcumin with QingDai is effective in preventing UC. Further, Dogan et al. [143] have also demonstrated that a Mediterranean diet rich in curcumin and resveratrol is effective in reducing inflammation, improving disease severity, and enhancing the patient’s quality of life. Thus, the current reported studies suggest that curcumin has beneficial effects in controlling IBD and UC. These studies also indicate that further development of standards for dosing and methods to improve bioavailability will provide more concrete clinical evidence for curcumin as a therapeutic option for these diseases (Figure 4).

## 6. Curcumin in Treating Sepsis

Sepsis is an impaired immunological response to an infection or injury in the body and can rapidly progress to result in organ dysfunction and failure [144,145]. Viral, bacterial, and fungal infections can give rise to sepsis and trigger a widespread systemic inflammatory response throughout the body. The dysregulation of immune events hinders the effective eradication of the infection, instead resulting in aggressive inflammation and potential damage to vital organs. Though it can develop in anyone, infants, adults over the age of 65, and those with chronic health disorders and lower immunity, for example, human immunodeficiency virus (HIV) patients, are at greater risk of experiencing sepsis [146]. There are more than 40 million cases of sepsis worldwide and 11 million sepsis-related deaths [147,148].

Sepsis treatment requires early management and begins with fluid resuscitation to stabilize blood volume and restore the central venous pressure to a normal range, ensuring that the body and vital organs are being well perfused and oxygenated. When fluid resuscitation fails to accomplish this goal, the use of vasopressor therapy can be utilized to raise blood pressure [149,150]. Generally, sepsis is triggered by severe bacterial infections. Prompt antibiotic treatment, ideally within 1 h of suspected sepsis, is very crucial for survival. In cases of septic shock, it has been shown that for every hour of antibiotic treatment delay, survival chances decrease significantly [151]. Despite novel treatment approaches and better symptom management, septic shock is still a major cause of mortality in ICU patients. Recent studies suggest that, beyond conventional antibiotic and anti-inflammatory treatment approaches for sepsis, curcumin has also shown promising outcomes in aiding therapeutic interventions [152]. Curcumin has been shown to reduce the production of NF-κB-mediated proinflammatory cytokines, which are critical for enhancing the inflammatory response that leads to sepsis [153]. Treatment with curcumin in the sepsis model of mice has demonstrated improved survival outcomes and reduced tissue damage [154]. Additionally, Zhao et al. [155] observed that CD4+ CD25- T cells underwent poor proliferation, indicating that prolonged activity of Tregs and an excessive increase in IL-10 can lead to immunoparalysis, thereby worsening the immune response and outcome. Increased expression of FOXP3 (Forkhead/winged helix transcription factor p3) was also observed in curcumin-treated septic mice. FOXP3 is integral in the regulation of Treg cell development and proliferation, and dysregulation of this transcription factor can lead to an impaired immune response [155]. This study concluded that curcumin is effective in decreasing widespread inflammation by upregulating Tregs and IL-10, which counteract inflammation in septic mice. They have also demonstrated that in septic mice, there was a decrease in plasma secretion of TNF-α and IL-6. These findings reinforce the notion that curcumin is a valuable compound for reducing inflammation, which holds promise in the treatment of sepsis [155].

Furthermore, mice treated with curcumin exhibited increased levels of CD4+ and CD25+ regulatory T cells (Tregs) and the interleukin-10 (IL-10) cytokine [156]. Tregs play a crucial role in maintaining a balanced immune response by suppressing the activity of CD4+ and CD25+ T cells. Tregs release IL-10 and TGF-β, inhibitory cytokines responsible for suppressing the immune responses.

In vitro studies have shown similar outcomes with curcumin treatment for sepsis. One such study investigated the role of a curcumin derivative known as FM0807 on lipopolysaccharide (LPS)-induced RAW 264.7 macrophages [157]. The cells pretreated with the curcumin derivative had decreased production of proinflammatory cytokines, including TNF-α, IL-6, and IL-β. FM0807 was also shown to be involved in inhibiting apoptosis through inhibiting the production of reactive oxygen species (ROS). The mechanism through which this derivative has such effects was also investigated. FM0807 was observed to inhibit the JNK pathway, which is activated in response to ROS [157]. They have also shown that FM0807 prevents the activation of p53, as well as caspase-3 and caspase-9, and thus protects against apoptotic cell death. These results demonstrate a promising avenue for sepsis treatment by utilizing curcumin derivatives to effectively decrease inflammation without inducing immunoparalysis. Similarly, curcumin-loaded exosomes derived from bone marrow stem cells (BMSCs-EXOCurcumin) have been found to reduce inflammation, oxidative stress, and apoptosis in LPS-induced HK2 kidney cells [158]. Furthermore, this study also indicates that in a mouse model of SA-AKI, BMSCs-EXOCurcumin improves kidney function by modulating the FTO/OXSR1 axis. Another study by Chen et al. [159] has shown that curcumin treatment significantly reduced lung injury, inflammation, oxidative stress, and ferroptosis markers in CLP-induced septic mice and LPS-stimulated Beas-2B cells. Specifically, curcumin decreased levels of IL-6, IL-1β, TNF-α, MDA, MPO, ROS, and Fe^2+^, restored the GSH levels, and regulated the expression of TXNIP, TRX-1, GPX4, and X-CT. They have further shown that these protective effects are reversed by the TRX-1 inhibitor PX-12, indicating that curcumin’s therapeutic action could be mediated through the TRX-1 pathway. Furthermore, Jiang et al. [160] have demonstrated that ceria-curcumin nanozyme particles (CeCH) exhibit both antioxidant (SOD- and CAT-like) and anti-ferroptotic activities, thereby protecting cardiomyocytes from cell death. They have also demonstrated that CeCH not only inhibits ferroptosis and promotes the polarization of anti-inflammatory M2 macrophages but also significantly reduces cardiac inflammation and improves heart function in the LPS-induced sepsis model. In another study, Qiu et al. [161] investigated the role of J147, a curcumin analog, in mitigating depressive-like behaviors caused by sepsis-associated encephalopathy (SAE). They found that J147 suppressed neuroinflammation in LPS-treated mice by downregulating proinflammatory cytokines (IL-6, IL-1β, and TNF-α), thereby inhibiting microglial activation through the blockade of the TLR4/NF-κB signaling pathway. These results suggest that J147 could be neuroprotective and reduce behavioral symptoms of depression, bodyweight loss, and mortality associated with SAE. Huang et al. [162] demonstrated that curcumin effectively reduced kidney injury and improved renal function in LPS-induced septic acute kidney injury (AKI) mice by lowering serum Scr, BUN, and cyclosporine C levels. They have shown that curcumin inhibited inflammation and apoptosis in both mouse kidney tissues and NRK cells by downregulating the lncRNA PVT1 and suppressing the JNK/NF-κB signaling pathway. On the other hand, Liu et al. [163] have shown that AI-44, a novel curcumin analog, binds to PRDX1, enhancing its interaction with pro-Caspase-1 and disrupting the assembly of the NLRP3 inflammasome. Furthermore, in LPS-induced endotoxemia models, AI-44 significantly reduced inflammation by preventing the NLRP3-mediated inflammatory response. Similarly, Gong et al. have shown that curcumin prevents LPS-induced NLRP3 activation and septic shock [164]. Furthermore, several additional studies have also suggested that curcumin prevents LPS-induced endotoxemia by down-regulating microRNA-155 [165], liver cirrhosis by regulating PCSK9 [166], liver failure by regulating the PI3K/AKT/NF-κB signaling [167], and cytotoxicity by regulating IRAK-MAPK signaling [168].

There is a lack of human studies regarding the potential therapeutic role of curcumin in sepsis; however, a clinical trial using nanocurcumin has shown promising results. In a randomized trial, 40 patients aged 18–55 with sepsis were administered 160 mg of curcumin supplement or a placebo supplement via a nasogastric tube for 10 days [169] (Table 1). Patients who received nanocurcumin treatment were observed to have significantly decreased levels of procalcitonin (PCT), IL-6, and TNF-*α*. PCT is known to be elevated in patients experiencing bacterial infections, surgery, trauma, and multi-organ failure. TNF-*α* activates the production of PCT, and thus PCT may be used as a marker for an inflammatory response.

Additionally, hs-CRP was also measured as a known biomarker for inflammation. No significant differences in hs-CRP levels were noted between the two groups. The length of hospital stays, mortality, and use of mechanical ventilation were also assessed among the patients, and results showed a remarkable decrease in the use of mechanical ventilation among patients treated with nanocurcumin. Another study by the same group also demonstrated that nanocurcumin supplementation (10 days) in ICU patients with sepsis significantly improved inflammatory markers (e.g., IL-6, IL-18, IL-1β, IL-10), endothelial function (ICAM-1, VCAM-1), and oxidative stress indices (MDA, Nrf-2, catalase, SOD, TAC) compared to placebo [170]. Furthermore, nanocurcumin significantly decreased the Sequential Organ Failure Assessment (SOFA) scores among patients treated with nanocurcumin. SOFA scores are widely used to assess morbidity in critically ill patients and are also used to determine the efficacy of therapeutic agents in treating sepsis [170]. The results of these clinical studies reinforce the notion that curcumin has significant therapeutic potential for treating septic patients. Further, a recent double-blind, randomized, controlled study by Alikiaii et al. [171] examined 66 ICU patients with sepsis who received curcumin-piperine supplements for 7 days. They have shown that this combination significantly improved inflammatory markers and hematologic parameters compared to the placebo group. Further, this intervention also reduced bilirubin levels, CRP, and ESR and helped preserve red blood cell indices. This study also indicates that the mortality rates were similar between groups. Thus, this study suggests that curcumin-piperine combination has beneficial anti-inflammatory and supportive effects in critically ill septic patients [118]. Similarly, Musso et al. [172] examined nutrient-induced inflammation and gut hormone responses in MASH patients with or without CKD and evaluated the impact of curcumin Meriva supplementation. They found that MASH patients with CKD had reduced postprandial GLP-2 responses and elevated NF-κB activation, which contributes to increased intestinal barrier dysfunction and systemic inflammation. Curcumin supplementation improved GLP-2 response and reduced markers of endotoxemia and inflammation over 72 weeks, suggesting curcumin may help control MASH-related CKD progression by enhancing gut integrity and reducing inflammation.

Sepsis and septic shock continue to be leading causes of death among critically ill patients in hospital settings. A multidisciplinary approach is necessary when treating septic patients to reduce rehospitalizations and the length of hospital stay. The initial approach to treatment remains antibiotic therapy, removal of infected tissue, restoration of venous pressure through fluid resuscitation, and, in some cases, mechanical ventilation. Curcumin is a compound that has been demonstrated to decrease extensive inflammation in sepsis effectively, and its therapeutic benefits are linked to its ability to counteract proinflammatory proteins, such as cytokines and transcription factors [173,174,175] (Figure 5). Through its effective reduction in inflammation, curcumin could prevent multi-organ damage and failure in sepsis (Figure 5). While additional research is necessary to establish the efficacy of curcumin in clinical studies, both in vitro and animal studies have demonstrated its potential in treating sepsis.

## 7. Curcumin in Treating Psoriasis and Atopic Dermatitis

Atopic dermatitis and psoriasis are common conditions that affect the skin and the immune system. Both are associated with chronic inflammation. Atopic dermatitis and psoriasis are the most common chronic inflammatory skin diseases [176]. In the USA, the prevalence of psoriasis is approximately 3%. The prevalence of atopic dermatitis in the USA is even higher, at 10.7% [177,178]. The prevalence of these disorders varies greatly with race and location, as well as with age. Atopic dermatitis and psoriasis are prevalent worldwide and have a significant impact on patients. The effect on skin’s appearance leads to lower self-confidence and a decreased desire for human interaction, which in turn affects individuals’ mental health and overall quality of life [179].

Psoriasis is a chronic, inflammatory disease identified by erythema and silvery scales on the body [170]. Chronic plaque psoriasis is the prevalent form of psoriasis, accounting for 90% of cases [180]. The onset of psoriasis is associated with T-cells, TNF-α, dendritic cells, and environmental triggers. T-cells are also implicated in the onset of atopic dermatitis, along with mast cells [180]. Atopic dermatitis typically begins in childhood and is also triggered by skin barrier dysfunction. In individuals with genetic predisposition, symptoms of psoriasis can be triggered by external factors such as infection or trauma [181]. Approximately 90% of patients with atopic dermatitis experience epidermal Staphylococcus aureus colonization, which contributes to inflammation and may lead to secondary infections [181,182].

Because atopic dermatitis and psoriasis are so prevalent and well-studied, many treatment options exist [183,184,185]. Treatment modalities depend on the severity of the disease. For mild psoriasis, topicals such as corticosteroids, retinoids, and vitamin D analogs are used. Phototherapy is used for moderate psoriasis, and systemic medications are used for severe psoriasis [185,186]. The primary treatment for atopic dermatitis involves repairing the skin’s barrier by focusing on topical treatments and less irritating skincare practices, such as using mild soap and avoiding hot water. Topical medications are the most common treatment for atopic dermatitis. However, typical side effects of topicals prescribed for atopic dermatitis include pruritus and a burning sensation. This highlights the need for novel therapies for atopic dermatitis.

Newer studies have improved the understanding of psoriasis and atopic dermatitis. Immune cells and cytokines, such as Th17, Th22, IL-31, and TSLP, have been identified in the development of atopic dermatitis, allowing for research into more specific and overall improved treatments [185,186]. There are several monoclonal antibody-based treatments currently being evaluated for atopic dermatitis. The main pathway identified in psoriasis is the TNF-α/IL-23/IL-17 axis, which provides three cytokine targets for treatment [187]. A few recent studies also suggest an impact of gut microbiota on psoriasis and dermatitis [188,189,190].

Turmeric has been used in traditional medicine for centuries in Asian countries to treat skin infections and injuries [191,192]. Recent studies have also reported on the use of curcumin to treat inflammatory skin complications such as psoriasis and atopic dermatitis [193,194] (Figure 6). Additionally, curcumin has been shown to have anti-inflammatory, antioxidant, anti-carcinogenic, anti-mutagenic, anti-coagulant, and anti-infective effects [194,195]. Because atopic dermatitis and psoriasis are known to be inflammatory conditions, studying anti-inflammatory compounds such as curcumin may lead to novel treatments for these skin conditions.

Recent studies suggest that curcumin is a novel treatment option for psoriasis [196]. The etiology of psoriasis involves the overproduction of Th1 and Th2-related cytokines, such as interleukin (IL)-23, IL-17, TNF-α, and IL-22. IL-6 is also implicated in psoriasis, as its expression is elevated in psoriatic lesions. Indeed, Cai et al. [197] have shown that curcumin treatment decreased Psoriasis Area and Severity Index (PASI) scores, lesional levels of IL-6 and TNF-α, and symptoms of psoriasis. They further demonstrated that curcumin reduced the levels of IL-17a, IL-22, IL-23, and TGF-β1, and increased the expression of the potent anti-inflammatory cytokine IL-10. A meta-analysis focused on clinical trials also concluded that curcumin improved PASI scores, both when used as monotherapy and in combination therapy [198,199]. Similarly, Xu et al. [200] have indicated that in psoriatic models, curcumin-loaded microneedles significantly improved skin lesions, lowered the PASI score, and suppressed key proinflammatory cytokines (TNF-α, IL-17, IL-22, and IL-23). Furthermore, Cai et al. [201] have also shown that curcumin, by regulating the gut microbiota, could prevent imiquimod-induced psoriasis in mice. In similar lines, Serini et al. [202] have also suggested that Curcumin-based solid lipid nanoparticles may inhibit psoriatic inflammation and hyperproliferation of keratinocytes in psoriatic lesions. Several other studies also suggest nanoparticle-mediated curcumin prevents psoriatic lesions in various experimental models [193,203,204,205,206]. A clinical study by Billa et al. [207] demonstrated that patients with moderate-to-severe psoriasis treated with acitretin plus nanocurcumin showed a significantly greater reduction in PASI scores compared to those on acitretin alone (Table 1). These results suggest that nanocurcumin is a safe and effective adjuvant that enhances psoriasis outcomes without affecting lipid profiles.

Atopic dermatitis is also caused by an imbalance in the body’s immune system. In early atopic dermatitis, there is an overproduction of Th2-related cytokines (IL-4, IL-5, IL-13, and IL-31), which switches to an overproduction of Th1-related cytokines (IL-1, IL-6, IL-12, IL-18, and TNF-α) in later phases of the disease [208]. Several studies have shown promising results in treating atopic dermatitis with curcumin [209]. Sharma et al. [210] have investigated how curcumin prevents ovalbumin-induced AD in a mouse model. They found that curcumin significantly reduces skin inflammation, epidermal thickening, and infiltration of inflammatory cells. Interestingly, curcumin also suppressed the key Th2 cytokines (IL-4, IL-5, IL-13, IL-31), upstream mediators (TSLP, IL-33), and signaling molecules (GATA-3, STAT6, p65-NF-κB). Similarly, Zhang et al. [211] have shown that curcumin compounds improve symptoms of atopic dermatitis via inhibiting the MAPK/NF-κB pathway. Further, Saini et al. [212] also evaluated the therapeutic potential of a tetrahydrocurcumin (THC) solid lipid nanoparticle gel for treating atopic dermatitis. They found that this gel has effectively improved skin hydration and enhanced the penetration of drugs.

Additionally, in 1-Chloro-2,4-dinitrobenzene (DNCB)-induced AD mice, the gel significantly reduced inflammatory markers (TNF-α and IL-6) and promoted complete skin healing. Similar results were also observed by Cassano et al. [213], Chen et al. [214], and Frei et al. [215]. In addition, Panahi et al. [216] have conducted a randomized clinical trial to demonstrate that curcumin supplementation (1 g/day for 4 weeks) significantly reduces chronic pruritus symptoms in Iranian veterans exposed to sulfur mustard (SM). They have shown that curcumin lowers the serum substance *p* and enhances antioxidant enzyme activity. These findings suggest curcumin as a safe, affordable, and effective natural treatment option for managing SM-induced chronic pruritus. Together, these studies show curcumin may be an effective alternative or addition to current treatments that focus on repairing the skin’s barrier (Figure 6). Although most studies on atopic dermatitis have been preclinical evaluations of curcumin or curcumin nano-formulations, additional clinical studies are required to identify its potential therapeutic use in atopic dermatitis.

**Table 1 ijms-26-09366-t001:** Curcumin clinical trials with different formulations and primary outcomes.

References	Design	Formulation	Dose	Duration	Comparator	Primary Outcome
Osteoarthritis						
[38]	Active-controlled trial (randomization/blinding; N = 139; ages 38–65)	Curcumin	1000–2000 mg/day	28 days	Diclofenac	Pain/function
[40]	RCT, double-blind, placebo-controlled, parallel, N = 19, age <80 years	Curcuminoids	1500 mg/day	6 weeks	Placebo	Pain/function
[44]	RCT, double-blind, placebo controlled, N = 30, ages 40–55	Curcumin	-	3 months	Placebo	Immunologic
[45]	Clinical study (topical); N = 60	SNE-PEG organogel	-	8 weeks	Placebo/standard care	Pain/function
[46]	Clinical study	Topical curcumin	10%	2 weeks	Placebo/diclofenac	Analgesia
Asthma/COPD						
[65]	Cross-sectional observational; N = 2478, ages > 55	Dietary turmeric/curcumin intake	≥1×/month	—	None	Lung function
[69]	Randomized clinical study, N = 77,	Curcumin capsules	500 mg × 2	30 days	Standard asthma therapy	Lung function
[83]	RCT, double-blind, controlled, Children and adolescents, ages 7–18 years	Powdered Curcuma longa roots	30 mg/kg/day	3- and 6-months follow-ups	Placebo	Control/reliever use
[84]	RCT, double-blind, placebo controlled; N = 60 COPD	Nanocurcumin	80 mg	3 months	Placebo/standard care	Lung function, cytokines
[82]	Randomized clinical study, N = 89, sulfur-mustard lung injury,	Curcuminoids + piperine	1500 mg/day	4 weeks	Placebo	Oxidative stress, COPD test
Atherosclerosis						
[111]	RCT, double-blind, placebo-controlled, N = 227 (T2DM), age > 35 years	Curcumin	250 mg × 2/day	12 months	Placebo	Arterial stiffness, lipids
[112]	RCT, double-blind, placebo-controlled, N = 64 (T2DM and mild to moderate CAD),	Nano-curcumin	80 mg/day	90 days	Placebo	Inflammation, lipoprotein
[113]	RCT, double-blind, placebo-controlled, age 20–85 years	Theracurmin^®^	90 mg × 2/day	24 weeks	Placebo	Atherogenic markers
IBD						
[136]	RCT, double-blind, placebo-controlled (add-on to mesalamine) N = 50	Curcumin + mesalamine	3 g/day	One month	Mesalamine + placebo	Remission, biomarkers
[138]	RCT, double-blind clinical trial; N = 70 (mild-moderate UC)	Curcumin + drug therapy	1500 mg/day	8 weeks	Placebo	Symptoms and inflammation
[141]	Multicenter, double-blind RCT pilot study, N = 69 (mild-moderate UC)	Curcumin + mesalamine	50 mg × 2/day	6 weeks–3 months	Mesalamine + placebo	Clinical remission
[140]	Multicenter, double-blind RCT, N = 30 (Crohn’s)	Theracurmin^®^	360 mg/day	12 weeks	Placebo/standard care	Endoscopic and clinical activity
[139]	RCT, double-blind, placebo-controlled, N = 58 (IBD), age > 18 years	Curcumin + piperine	1000 mg/day	12 weeks	Placebo	Muscle status
Sepsis						
[169]	Pilot randomized trial, N = 40 ICU, ages 18–55 years	Nanocurcumin	160 mg/day	10 days	Placebo	Inflammation, organ failure
[170]	RCT, double-blind, placebo-controlled, N = 40 ICU	Nanocurcumin	160 mg × 2/day	10 days	Placebo	Inflammation, oxidative stress
[171]	Double-blind RCT, N = 66 ICU, ages 20–75 years	Curcumin + piperine	500 mg/day	7 days	Placebo	Inflammation, hematology
[172]	Prospective study, N = 52 (MASH + CKD)	Curcumin Meriva^®^	2 g/day	72 weeks	None	Gut barrier, inflammation
Psoriasis						
[207]	RCT, double-blind, placebo-controlled, moderate to severe psoriasis	Nanocurcumin + acitretin	3 g/day	12 weeks	Acitretin	PASI
[216]	RCT, double-blind, placebo-controlled, N = 96 (sulfur mustard -chronic pruritus), ages 37–59 years	Curcumin	1 g/day	4 weeks	Placebo	Pruritus, biomarkers

## 8. Safety Profile and Adverse Events with Curcumin Use

Several clinical studies were performed using curcumin and curcumin nanoformulations. However, safe use of curcumin has been well established; its adverse events (AEs) were inconsistent and minimally reported across clinical trials. In one of the strongest RCTs, Banerjee et al. [141] have found that there were no additional adverse effects when curcumin was administered with mesalamine in patients with ulcerative colitis. Similarly, ICU trials of nanocurcumin (160 mg/day for 10 days) demonstrated improvements in inflammatory and endothelial markers and reduced SOFA scores without excess mortality compared to placebo. However, in these studies, the AE reporting was limited [169,170,171]. In osteoarthritis studies, curcumin provided significant symptom improvements and reduced NSAID use [38,40,43,44]. Similarly, topical formulations such as curcumin gels and organogels were shown to be effective and well-tolerated [45,46].

Further, in psoriasis, nanocurcumin combined with acitretin has been shown to improve PASI scores without affecting lipid profiles [207]. Asthma and COPD studies also showed curcumin’s clinical benefits [65,69,83,84,113]. Similarly, the co-administration of piperine with curcumin has been shown to increase the curcumin bioavailability by up to 2000% [79,80,81,82], raising theoretical risks for drugs with narrow therapeutic windows. However, these studies have not clearly documented bleeding events, INR alterations, or cytochrome-P450-mediated drug–drug interactions. Further, the reported doses varied widely across trials with short-term and long-term exposure to curcumin. For example, native oral curcumin at ~1000–2000 mg/day [38], nanocurcumin at 160 mg/day in ICU patients [169], curcumin with piperine for 7 days (500 mg curcuminoids and 5 mg piperine) [171], and Theracurmin^®^ (360 mg/day) over 12–24 weeks [113,140]. Long-term exposure data remain limited. The longest trial, a 12-month cardiometabolic study, demonstrated vascular benefits of curcumin (250 mg of curcuminoids) and suggested that curcumin is safe to use at a dose of 1500 mg/day [111]. Further, the therapeutic efficacy of curcumin is based on the doses of curcumin in various formulations and treatment duration (Table 2). Overall, curcumin appears to be safe and well-tolerated in short-term interventions, but small sample sizes, brief study durations, and heterogeneous formulations limit confidence. Furthermore, improved AE reporting is necessary to establish a more comprehensive safety profile. Additionally, long-term clinical trials including large populations and rigorous AE monitoring and thorough assessment of drug–drug interactions are still required to address inflammatory complications.

**Table 2 ijms-26-09366-t002:** The therapeutic efficacy of curcumin is based on the doses of curcumin in various formulations and treatment duration.

Formulation	Indication(s)	Typical Dose Range	Duration Range	References
Plain oral curcumin	OA, UC/CD, Asthma, Psoriasis	~1000–2000 mg/day (OA)	4–12 weeks typical; up to 12 months	[38,40,43,46,65,69,135,138,141,197]
Curcumin + piperine	COPD (sulfur-mustard lung injury), IBD (muscle wasting), ICU sepsis	Dose 500 mg/day (5 mg piperine co-administered)	7 days to multi-week	[82,139,171]
Nanocurcumin	ICU sepsis, COPD, Psoriasis	160 mg/day (ICU)	10 days (ICU) to multi-week	[84,169,170,207]
Theracurmin^®^	COPD, Crohn’s disease	360 mg/day	12–24 weeks	[113,140]
Topical (organogel/cream)	Knee OA, Atopic Dermatitis	10%	2 weeks	[45,46,212,213,214]
Analogs (J147, AI-44)	Sepsis models (preclinical, SAE, endotoxemia)	-	-	[160,161,163]

The highest-quality randomized clinical trials (RCTs) that include the use of risk of bias 2 (RoB 2) are the UC trial by Banerjee et al. [141], the OA RDBPC trial by Panahi et al. [40], and the pediatric asthma trial by Manarin et al. [83]. All these studies have demonstrated a low to moderate risk of bias, with appropriate randomization, blinding, and well-defined outcomes. Other RCTs, such as the sepsis nanocurcumin studies [169,170] and the COPD Theracurmin^®^ study [113], demonstrated biomarker improvements but were limited by short duration and modest sample sizes. Further, another OA trial by Shep et al. [38] lacked sufficient information on randomization and blinding, placing it at high risk of bias. In addition, the Newcastle–Ottawa Scale (NOS) was applied to the COPD/asthma dietary study by Ng et al. [65], which scored moderately due to a strong sample size but limitations in exposure measurement and residual confounding. Thus, only a handful of curcumin studies qualify as high-quality RCTs with low RoB 2, while many others fall into the moderate or high-risk categories. Furthermore, the RoB 2 and NOS assessment tools support moderate confidence in the short-term effects of curcumin. However, further, larger and better clinical trials are required to understand the confidence levels better. With these improvements, the translation of promising preclinical mechanisms into clinical settings will significantly improve.

## 9. Conclusions and Future Perspectives

Without a doubt, curcumin, the principal curcuminoid derived from the roots of *Curcuma longa* (turmeric), has evolved as a potential therapeutic agent for a wide range of inflammatory diseases. For several thousand years, it has been used as a dietary supplement and natural antibiotic in many Asian countries. Further, extensive research studies over the past two decades have demonstrated the pleotropic functions of this agent. They include anti-inflammatory, antioxidant, antimicrobial, and anticancer properties. Specifically, curcumin has been shown to be beneficial in several preclinical and clinical models of osteoarthritis, asthma, chronic obstructive pulmonary disease (COPD), atherosclerosis, irritable bowel disease (IBD), sepsis, atopic dermatitis, and psoriasis. For instance, in the treatment of osteoarthritis, curcumin has been shown to reduce pain. For asthma and COPD, curcumin suppressed the production of proinflammatory cytokines and decreased pulmonary edema and airway constriction. Furthermore, in atherosclerosis, curcumin reduces plaque formation, inflammation, and artery stiffness and improves lipid profiles. Curcumin enhances cellular antioxidant defense, inflammation, and tissue damage in patients with IBD. Similarly, in sepsis, curcumin can decrease widespread inflammation and greatly reduce the need for mechanical ventilation and the Sequential Organ Failure Assessment score. Thus, multiple studies have shown beneficial effects of curcumin use in OA, asthma, COPD, atherosclerosis, IBD, sepsis, atopic dermatitis, and psoriasis. However, further research is needed to understand its molecular mechanisms of action and to determine its efficacy during long-term use.

Most of the recent studies also indicate that curcumin can regulate multiple molecular targets simultaneously. It can inhibit key inflammatory mediators and signaling pathways such as the NF-κB-mediated signalosome pathway and the NLRP3-mediated inflammasome pathway. These pathways activate various inflammatory and immune responses by activating cytokines such as IL-1β, IL-6, IFN-γ, TNF-α, and IL-18, and by regulating the activation of various kinase cascades mediated by PKC, MAPK, AMPK, and others. Thus, the regulation of multiple pathways by curcumin makes curcumin particularly suitable for treating complex, multifactorial diseases where inflammation plays a central role (Table 3). Furthermore, its favorable safety profile, even at relatively high doses, is beneficial as a therapeutic agent alone or as an adjuvant to conventional treatment options. Additionally, further identification of specific disease biomarkers and understanding the interactions of curcumin with these biomarkers is crucial in developing effective therapeutic strategies. Furthermore, the interaction of curcumin with gut microbiota may also aid in the development of new treatment options. Moreover, studies using recent omics technologies, such as transcriptomics and metabolomics, could also help elucidate the mechanism of action of curcumin and identify potential biomarkers.

**Table 3 ijms-26-09366-t003:** Mechanistic vs. clinical biomarker evidence for curcumin’s therapeutic use in preclinical and clinical studies.

Pathway/Mechanism	Preclinical Evidence	Human Biomarker Evidence (Clinical Trials)	Limits of Translation from Preclinical to Clinical
NF-κB/AP-1 signaling	Inhibition of NF-κB, AP-1, and downstream cytokines across multiple models (OA chondrocytes, asthma, sepsis, IBD) [18,19,20,21,39,70,122].	Decreased l inflammatory markers such as IL-6, TNF-α, and CRP in OA, COPD, UC/IBD, and ICU sepsis. [40,43,82,84,136,138,141,169,170,171].	Clinical data align with NF-κB inhibition and inhibition of inflammatory cytokines. Other upstream signals of NF-kB were not directly measured.
NLRP3 inflammasome	Inhibition of NLRP3 activation, inhibition of IL-1β, and caspase-1 inhibition in DSS-colitis, sepsis, endotoxemia models [24,129,163,164].	Decreased IL-1β observed in ICU sepsis (nanocurcumin) [170].	Mostly preclinical studies; Il-1b levels in patients. Additional clinical data is required for direct inflammasome readouts (caspase-1 activity, and complex proteins).
MAPK/JNK/p38 pathways	Inhibits MAPK phosphorylation and JNK-mediated apoptosis in cells and sepsis models [39,122,157].	Decreased CRP and TNF-α, improved WOMAC/VAS scores in OA [40].	Clinical biomarkers do not capture MAPK activity; human confirmation studies are still needed.
Oxidative stress/antioxidant defense	Decreased ROS, MDA, MPO; increased Nrf-2, catalase, SOD, TAC in cell and animal models [157,159,160].	In ICU sepsis trials, decreased MDA, and increased Nrf-2, catalase, SOD, TAC with nanocurcumin [170]; in COPD, improved oxidative stress indices [82].	Stronger translational connection. Both preclinical and human subject data support antioxidant effects.
Endothelial activation	Decreased ICAM-1 and VCAM-1, improved vascular function in atherosclerosis [106,107,108].	In ICU sepsis, decreased ICAM-1 and VCAM-1 [170]; In atherosclerosis, decreased arterial stiffness, Lp(a), and AT-LDL [111,112,113].	Both preclinical and human trials confirm endothelial biomarker modulation.
Ferroptosis	Curcumin and CeCH analogs decrease ferroptosis markers, restore GPX4, and prevent cardiac/kidney injury in sepsis models [159,160].	No clinical ferroptosis biomarkers reported.	Translation gap is observed. No human data on GPX4, lipid peroxides, or ferroptosis-specific panels.
Immune cell modulation	Inhibition of Th17 cells, increased Tregs, modulation of B/T follicular helper cells in IBD models [123,124,125,126,127].	In UC trials, CRP/IL-6 decreased and improved remission rates [136,141].	Clinical trials are limited to measuring the cytokines, no direct immune cell modulation, and Th17/Treg readouts in clinical settings.

However, one of the significant drawbacks of curcumin use in clinical applications is due to its poor oral bioavailability, rapid metabolism, and low systemic absorption. To overcome these limitations, significant advances have been made recently in the development of nanoformulations aimed at enhancing their pharmacokinetics and therapeutic efficacy (Figure 7). Several studies have developed nanotechnology-based delivery systems, such as nanoparticles, solid lipid nanoparticles, liposomes, nanoemulsions, and phytosomes, which have increased solubility, stability, and bioavailability (Table 4). For example, several studies have shown that curcumin-loaded nanoparticles can promote sustained release and facilitate enhanced cellular uptake and tissue distribution. Furthermore, recent clinical trials using nano-formulated curcumin have demonstrated that these formulations have improved plasma concentrations and therapeutic responses in patients with inflammatory conditions. These studies have also shown significant clinical strategies using these nanoparticles. Moreover, additional studies examining its synergistic interactions with established anti-inflammatory drugs and immunosuppressants could enable us to lower the dose levels and reduce the unwanted side effects of anti-inflammatory drugs, such as NSAIDs. Further, several promising RCTs suggest the therapeutic efficacy of curcumin; however, no curcumin formulation is currently FDA-approved for treating inflammatory diseases. This could be due to inconsistent trial quality, lack of standardized dosing, and absence of validated pharmacokinetic data. One should also consider the manufacturing challenges of nano-curcumin or other formulations, including their stability, scalability, and quality control, which also act as roadblocks to further development. Additional cost–benefit analysis studies are required to compare various curcumin formulations with standard anti-inflammatory drugs.

**Table 4 ijms-26-09366-t004:** Curcumin delivery systems and their use in various preclinical and clinical studies.

Disease	Delivery System(s)	Major Effects	References
Osteoarthritis (OA)	Oral capsules/extracts	Comparable efficacy to NSAIDs (diclofenac, naproxen) and reduced pain (VAS, KOOS, WOMAC).	[38,40]
	Self-nano-emulsifying PEG organogel (SNE-POG)	Improved absorption and efficacy in knee OA.	[45]
	Topical curcumin formulations	Analgesic effect and reduced knee pain.	[46]
	PGA-curcumin combination (dogs)	Reduced meloxicam dosage (~25%); effective pain control.	[41]
	Nanocurcumin/Phytosomes	Improved bioavailability, reduced IL-1β, NO, PGE2, and MMP-3 in chondrocytes.	[39,43,44]
Asthma and COPD	Oral curcumin capsules/extracts	Reduced airway inflammation, NF-κB inhibition and improved FEV1/FVC.	[65,66,70,71]
	Nanocurcumin	Improved lung function in COPD and decreased IL-6.	[84]
	Curcumin + Piperine	Improved bioavailability, oxidative stress and COPD indices.	[79,80,81,82]
	Theracurmin^®^ (nano-curcumin)	Reduced AT-LDL in COPD and improved vascular health.	[113]
Atherosclerosis	Nano-curcumin	Reduced hs-CRP, lipoprotein A and improved lipid profiles in CAD.	[112]
	Theracurmin^®^	Reduced AT-LDL and improved cardiovascular risk markers.	[113]
	Nanoparticles/liposomes	Modulated macrophage polarization (M1→M2) and reduced plaque and cytokines (TNF-α, IL-6).	[101,102,103,104,105,108]
Inflammatory Bowel Disease (IBD)	Oral curcumin + mesalamine	Induced remission in UC and reduced relapse.	[136,141]
	Nano-curcumin/Theracurmin	Improved mucosal healing and reduced NF-κB activity.	[140]
	Curcumin + Piperine	Enhanced absorption, reduced inflammation and muscle depletion.	[139]
	Curcumin analogs (C66, derivatives)	Blocked JNK/NF-κB and NLRP3 pathways; reduced colitis severity.	[128,129]
Sepsis	Nanocurcumin (oral/NG tube)	Reduced IL-6, TNF-α, and PCT, and improved SOFA scores.	[169,170]
	Curcumin + Piperine	Improved inflammatory and hematologic markers in ICU patients.	[171]
	Curcumin-loaded exosomes (BMSC-ExoCurcumin)	Reduced oxidative stress and kidney injury in septic models.	[158]
	Ceria-curcumin nanozymes (CeCH)	Antioxidant and anti-ferroptotic, reduced heart and kidney inflammation.	[160]
	Curcumin analogs (FM0807, J147, AI-44)	Inhibited NF-κB, JNK/MAPK, and NLRP3 pathways, and improved survival.	[157,161,163,164]
Psoriasis	Oral nanocurcumin	Significant PASI reduction	[207]
	Curcumin-loaded microneedles	Improved lesions and lowered TNF-α, IL-17, IL-22.	[200]
	Solid lipid nanoparticles	Reduced keratinocyte proliferation and suppressed psoriatic inflammation.	[202]
	Curcumin topical gels	Reduced lesional cytokines (IL-6, TNF-α).	[197,198]
	Curcumin-based microbiota modulation	Prevented imiquimod-induced psoriasis in mice.	[201]
Atopic Dermatitis	Tetrahydrocurcumin (THC) SLN gel	Improved hydration and reduced TNF-α, IL-6; promoted healing.	[212]
	Nanoformulations (lipid nanoparticles, gels)	Reduced inflammatory cell infiltration and prevented thickening.	[209,210,211,213,214,215]
	Oral curcumin supplementation	Reduced chronic pruritus and lowered serum substance *p*.	[216]

Although recent studies suggest the significance of curcumin in the therapy of inflammatory diseases is promising, conclusions must remain appropriately tempered based on the study type and formulation. Additionally, most available studies are highly heterogeneous with respect to disease type, trial design, formulations used, dosing regimens, and outcome measures. Furthermore, many of these trials are small, short-term, and often lack rigorous blinding or standardized comparators, which limits their generalizability. Thus, most importantly, current findings should not be interpreted as establishing equivalence between curcumin and conventional therapies (NSAIDs, corticosteroids, biologics). Additional head-to-head comparison studies and large-scale randomized controlled trials are still needed. As such, some reported studies suggest that curcumin is better considered as a potential adjunct rather than a proven replacement therapy for inflammatory complications.

Moreover, further large-scale, double-blind, placebo-controlled clinical trials are necessary to confirm the long-term safety and optimal dosing of these nanoformulations. Further, the dosage should be adjusted based on the nanoformulation as well as the type of inflammatory complication. Although over-the-counter curcumin is available in all commercial sources, excessive intake as supplements should be cautioned, and a physician consultation is needed before taking any over-the-counter food supplements. Thus, recent clinical studies using innovative drug delivery technologies indicate that curcumin may soon transition from a traditional remedy to a scientifically validated therapeutic option for managing various inflammatory conditions.

## Figures and Tables

**Figure 1 ijms-26-09366-f001:**
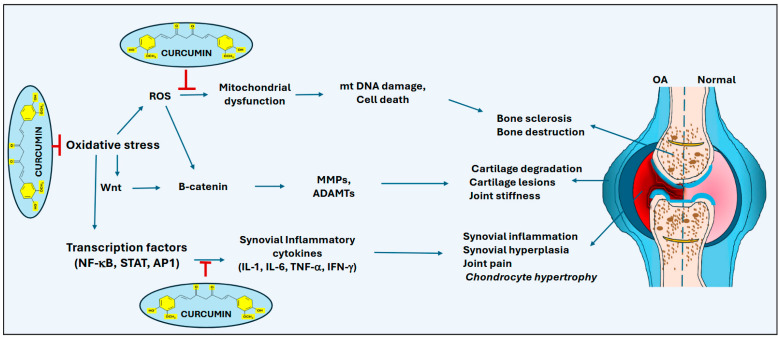
Curcumin protects from osteoarthritis (OA) progression by reducing oxidative stress and inflammation. Generally, reactive oxygen species (ROS) trigger mitochondrial dysfunction, DNA damage, and activation of transcription factors. These processes in turn lead to cartilage degradation, bone destruction, synovial inflammation, and joint pain. Curcumin blocks ROS production and inhibits inflammatory signaling pathways, thereby reducing cartilage lesions, joint stiffness, and synovial hyperplasia.

**Figure 2 ijms-26-09366-f002:**
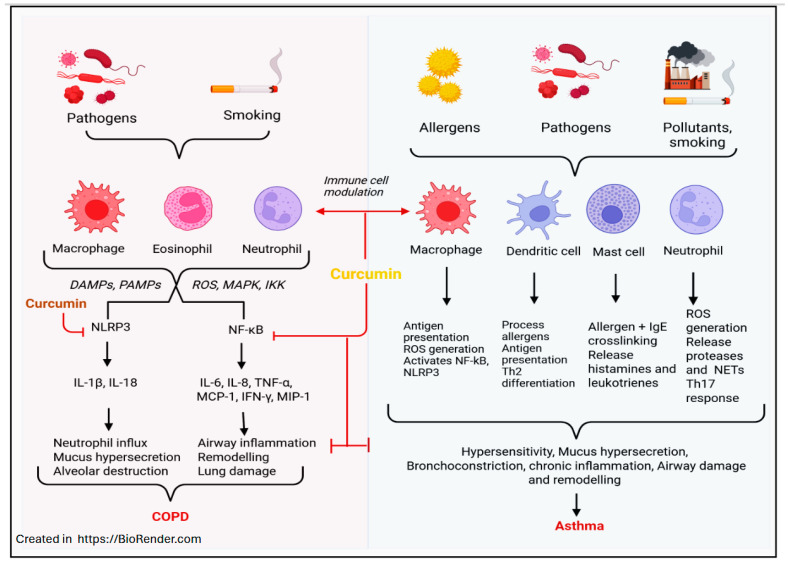
Curcumin modulates immune pathways to prevent COPD and Asthma. In COPD, macrophages, eosinophils, and neutrophils could trigger NLRP3 inflammasome and NF-κB signalosome signaling. The increase in immune and inflammatory responses could cause neutrophil influx, airway inflammation, mucus hypersecretion, and lung damage. In asthma, immune cells could activate Th2 and Th17 responses, leading to hypersensitivity, airway remodeling, and chronic inflammation. Curcumin inhibits these pathways by modulating immune cell activation and blocking NF-κB and NLRP3 signaling, resulting in reduced inflammation and tissue damage in both COPD and asthma. The image was created by using BioRender.com.

**Figure 3 ijms-26-09366-f003:**
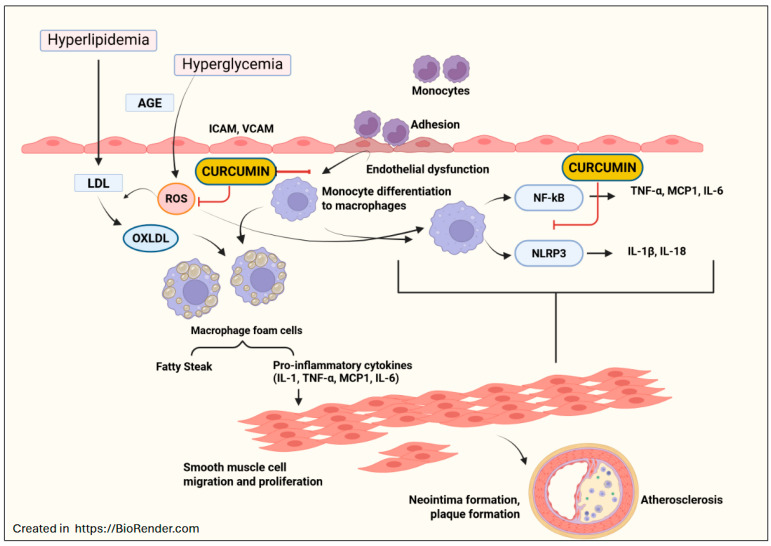
Role of curcumin in the prevention of atherosclerosis. Generally, hyperlipidemia and hyperglycemia are the major causes of atherosclerosis through oxidative stress, endothelial dysfunction, and inflammatory response. LDL oxidation and ROS have been shown to promote macrophage foam cell formation, fatty streaks, and the release of proinflammatory cytokines. These factors stimulate smooth muscle cell migration, proliferation, and plaque formation. Furthermore, monocyte adhesion and activation of NF-κB and NLRP3 can amplify cytokine production and vascular injury. Curcumin has been shown to block ROS, NF-κB, and NLRP3 pathways, and thus suppresses inflammation, foam cell formation, and atherosclerotic plaque development. The image was created by using BioRender.com.

**Figure 4 ijms-26-09366-f004:**
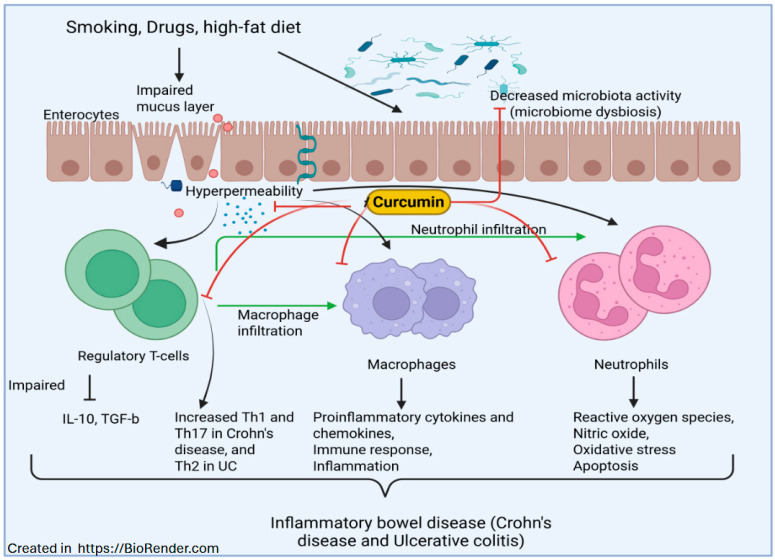
Prevention of intestinal Inflammation in IBD by curcumin. Oxidants such as smoking, drug use, and high-fat diets can disrupt the gut barrier by impairing the mucus layer, disturbing tight junctions, and reducing microbiota activity. These changes drive macrophage and neutrophil infiltration, excessive release of proinflammatory cytokines, and lead to Th1/Th17 responses in Crohn’s disease and Th2 responses in ulcerative colitis. Curcumin has been shown to regulate these processes by preserving microbiota balance, reducing immune cell activation, and preventing IBD. The image was created by using BioRender.com.

**Figure 5 ijms-26-09366-f005:**
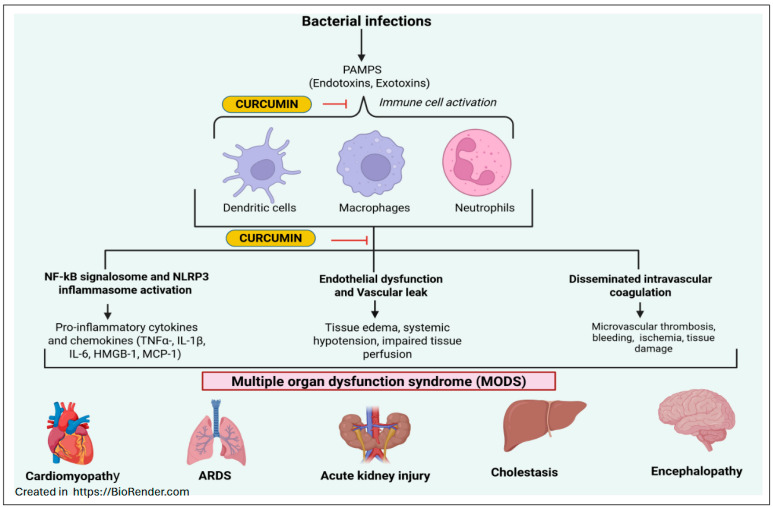
Curcumin prevents Multiple Organ Dysfunction Syndrome in sepsis. Bacterial infections cause sepsis by increasing immune and inflammatory responses. Uncontrolled activation of these responses leads to the increased production of proinflammatory cytokines and chemokines, resulting in endothelial dysfunction and disseminated intravascular coagulation. These processes cause tissue damage and dysfunction, leading to multiple organ dysfunction syndromes in sepsis. Curcumin could inhibit immune cell activation, NF-κB/NLRP3 signaling, and vascular dysfunction, thereby reducing systemic inflammation and organ dysfunction. The image was created by using BioRender.com.

**Figure 6 ijms-26-09366-f006:**
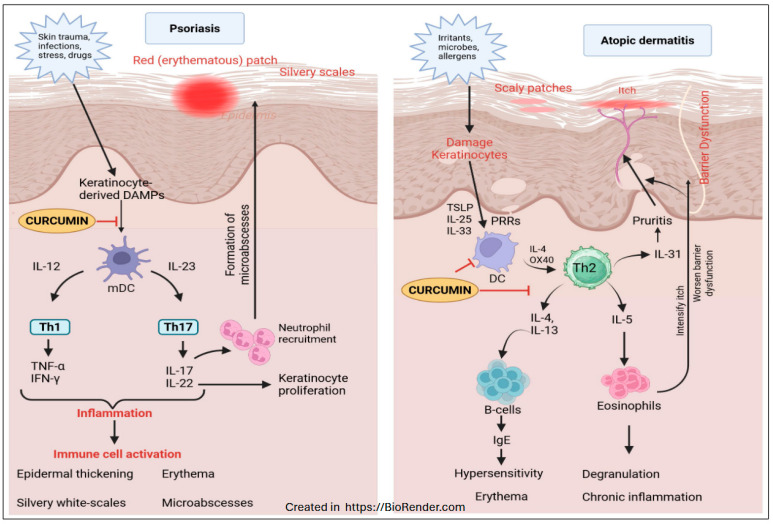
Significance of curcumin use in the prevention of psoriasis and atopic dermatitis. Curcumin has been shown to prevent psoriasis and atopic dermatitis due to aberrant immune responses in the skin. In psoriasis, keratinocyte-derived DAMPs activate dendritic cells to release IL-12 and IL-23, cause Th1/Th17 activation and cytokine release. Increased immune activation could cause epidermal thickening, erythema, and silvery scales. In atopic dermatitis, keratinocyte damage activates dendritic cells and Th2 pathways, leading to IgE-mediated hypersensitivity, eosinophil degranulation, and barrier dysfunction with pruritus. Curcumin could suppress dendritic cell activation and downstream cytokine responses in both conditions, reducing inflammation, hypersensitivity, and skin barrier damage. The image was created by using BioRender.com.

**Figure 7 ijms-26-09366-f007:**
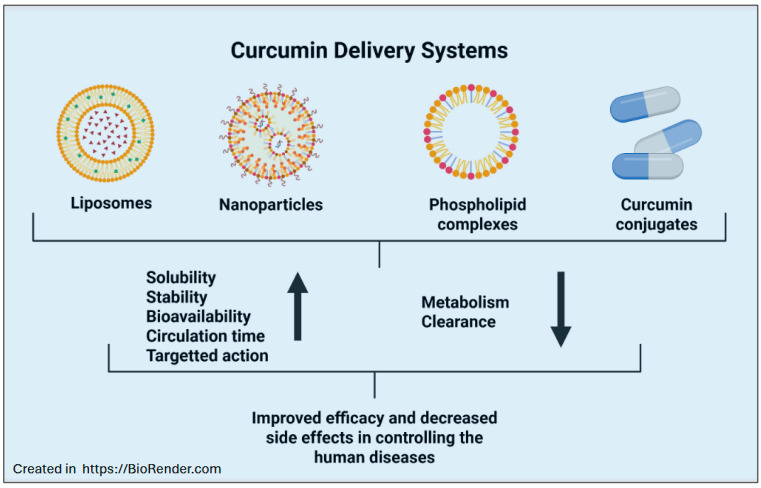
Curcumin delivery systems. The use of various nano-formulations, including liposomes, nanoparticles, phospholipid complexes, and curcumin conjugates, has been shown to enhance solubility, stability, bioavailability, circulation time, and targeted action. Furthermore, by strengthening these properties and reducing metabolism and clearance, novel delivery techniques could improve the therapeutic efficacy of curcumin and minimize its side effects. The image was created by using BioRender.com.

## Data Availability

No new data were created or analyzed in this study. Data sharing is not applicable to this article.

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
