# Peer review of "Curcumin in Inflammatory Complications: Therapeutic Applications and Clinical Evidence"

_ijms, 2025, doi:10.3390/ijms26199366_

Round 1
Reviewer 1 Report
Comments and Suggestions for Authors
This narrative review summarizes preclinical and clinical evidence that curcumin (and nano-/analog formulations) shows anti-inflammatory activity across osteoarthritis, asthma/COPD, atherosclerosis, IBD, sepsis, psoriasis and atopic dermatitis, but highlights low oral bioavailability and the need for larger, better-designed clinical trials.
However, there are several issues, which must be solved before it is considered for publication. If the following problems are well-addressed, I believe that the paper can be published.
1. The Methods' description is vague: “selected from PubMed searches … past decade” with few exceptions, but there is no explicit search strategy, search dates, keywords, number of records retrieved, screening criteria, or PRISMA flow. This undermines reproducibility and raises selection bias concerns. Revise to include full search strategy (databases, complete search strings, date range, language limits), screening process, and a PRISMA flow diagram or explicit statement why this is a narrative rather than systematic review.
2. Multiple clinical results are cited (e.g., OA trials, IBD remission, sepsis nanocurcumin), but the manuscript presents findings largely as fact without appraising trial quality (randomization, blinding, sample sizes, endpoints, intention-to-treat, attrition, funding/conflicts). Add a table of clinical trials (author, year, population, design, N, dose/formulation, duration, key outcomes, safety signals) and perform a formal quality assessment (e.g., Cochrane Risk of Bias 2 or at least Jadad/CONSORT considerations). This is essential before making clinical recommendations.
3. Statements implying curcumin “reduces pain” or “induces remission” do not acknowledge heterogeneity (doses ranging widely, different formulations, small N, variable comparators). The conclusions should be tempered and explicitly discuss heterogeneity and imprecision, and avoid implying equivalence to standard therapies without high-quality head-to-head data. Provide effect sizes or ranges where possible (e.g., mean WOMAC changes, PASI reductions) rather than only directional statements.
4. The manuscript asserts a “favorable safety profile” but cites limited short-term trials. Discuss known adverse effects, drug interactions (e.g., with anticoagulants, CYP interactions), maximum studied doses/durations, and gaps (lack of large long-term safety studies). Require explicit statements on safety evidence and limitations.
5. The review mixes native curcumin, curcuminoid extracts, Theracurmin®, nanocurcumin, curcumin + piperine, analogs (J147, AI-44), topical gels, etc., without systematically separating results by formulation. Since bioavailability and pharmacokinetics drive efficacy, reorganize sections to distinguish plain curcumin trials from enhanced formulations and to present dose/concentration ranges and pharmacokinetic data (Cmax, AUC when available).
6. Many molecular mechanisms (NF-κB, NLRP3, MAPK, ferroptosis, etc.) are described, largely from preclinical studies; the manuscript should clearly separate preclinical mechanistic evidence from human biological/biomarker data, and discuss which mechanisms have clinical biomarker support. Also add discussion of species/dose scaling and limits of preclinical-to-clinical translation.
7. Please add an explicit search strategy (databases and full search strings), dates searched, PRISMA flow diagram or rationale for narrative review format.
8. Please for each disease create a table listing trials (author, year, design, N, formulation, dose, duration, comparator, primary outcome, outcome measure and numeric result, adverse events). Highlight trials with high/low risk of bias.
9. Please apply an established tool (RoB2, Newcastle-Ottawa, etc.) and briefly summarize quality; discuss how quality affects confidence in conclusions.
10. Please reorganize results to clearly distinguish plain curcumin vs enhanced formulations, and list pharmacokinetic improvements (relative AUC/Cmax) where available.
11. Please add a dedicated safety subsection summarizing AEs, known interaction risks, and limits of current safety data.
Author Response
This narrative review summarizes preclinical and clinical evidence that curcumin (and nano-/analog formulations) shows anti-inflammatory activity across osteoarthritis, asthma/COPD, atherosclerosis, IBD, sepsis, psoriasis and atopic dermatitis, but highlights low oral bioavailability and the need for larger, better-designed clinical trials.
However, there are several issues, which must be solved before it is considered for publication. If the following problems are well-addressed, I believe that the paper can be published.
- The Methods' description is vague: “selected from PubMed searches … past decade” with few exceptions, but there is no explicit search strategy, search dates, keywords, number of records retrieved, screening criteria, or PRISMA flow. This undermines reproducibility and raises selection bias concerns. Revise to include full search strategy (databases, complete search strings, date range, language limits), screening process, and a PRISMA flow diagram or explicit statement why this is a narrative rather than systematic review.
As suggested, we have included an explicit statement about why this article is narrative rather than systematic. The following sentences have been included in the revised manuscript. This article is a narrative review rather than a systematic review because, although we synthesized evidence from multiple peer-reviewed studies on curcumin, this article does not follow a standard protocol for study identification, selection, or appraisal. Further, the literature search was limited to PubMed using selected keywords and restricted to certain years and inflammatory diseases only without a structured process such as PRISMA flow diagrams, risk-of-bias assessments, or meta-analysis. Through this article, we intend to provide a descriptive overview and thematic discussion on curcumin’s use in certain focused areas of inflammatory complications.
- Multiple clinical results are cited (e.g., OA trials, IBD remission, sepsis nanocurcumin), but the manuscript presents findings largely as fact without appraising trial quality (randomization, blinding, sample sizes, endpoints, intention-to-treat, attrition, funding/conflicts). Add a table of clinical trials (author, year, population, design, N, dose/formulation, duration, key outcomes, safety signals) and perform a formal quality assessment (e.g., Cochrane Risk of Bias 2 or at least Jadad/CONSORT considerations). This is essential before making clinical recommendations.
As suggested, we have included a table, (now Table-1) in the revised manuscript by including all the complications at one place and with above information.
- Statements implying curcumin “reduces pain” or “induces remission” do not acknowledge heterogeneity (doses ranging widely, different formulations, small N, variable comparators). The conclusions should be tempered and explicitly discuss heterogeneity and imprecision, and avoid implying equivalence to standard therapies without high-quality head-to-head data. Provide effect sizes or ranges where possible (e.g., mean WOMAC changes, PASI reductions) rather than only directional statements.
As suggested, we have rewarded the statements and discussed the heterogeneity. (please see conclusions, highlighted section)
- The manuscript asserts a “favorable safety profile” but cites limited short-term trials. Discuss known adverse effects, drug interactions (e.g., with anticoagulants, CYP interactions), maximum studied doses/durations, and gaps (lack of large long-term safety studies). Require explicit statements on safety evidence and limitations.
As suggested, we have included this information and also as suggested in the comment 11, we have included a separate side heading on “Safety profile and adverse effects of curcumin use”.
- The review mixes native curcumin, curcuminoid extracts, Theracurmin®, nanocurcumin, curcumin + piperine, analogs (J147, AI-44), topical gels, etc., without systematically separating results by formulation. Since bioavailability and pharmacokinetics drive efficacy, reorganize sections to distinguish plain curcumin trials from enhanced formulations and to present dose/concentration ranges and pharmacokinetic data (Cmax, AUC when available).
As suggested, we have included this information. However, we cannot reorganize sections as we are discussing multiple inflammatory complications in one review article. Further, as suggested in the point we have included additional table showing plain curcumin trails vs enhanced formulations at one place. Please see now Table 2.
- Many molecular mechanisms (NF-κB, NLRP3, MAPK, ferroptosis, etc.) are described, largely from preclinical studies; the manuscript should clearly separate preclinical mechanistic evidence from human biological/biomarker data, and discuss which mechanisms have clinical biomarker support. Also add discussion of species/dose scaling and limits of preclinical-to-clinical translation.
As suggested, we have included this information in additional table (now Table 3).
- Please add an explicit search strategy (databases and full search strings), dates searched, PRISMA flow diagram or rationale for narrative review format.
As suggested in this comment as well as comment number 1, we have indicated that this is a narrative review and therefore explicit search strategy was not included. However, general search strategy was included.
- Please for each disease create a table listing trials (author, year, design, N, formulation, dose, duration, comparator, primary outcome, outcome measure and numeric result, adverse events). Highlight trials with high/low risk of bias.
As suggested, in comment 2 and this one, we have included additional table showing the information requested (please see Table 1).
- Please apply an established tool (RoB2, Newcastle-Ottawa, etc.) and briefly summarize quality; discuss how quality affects confidence in conclusions.
As suggested, we have briefly summarized the information. Please see section 8, last paragraph.
- Please reorganize results to clearly distinguish plain curcumin vs enhanced formulations, and list pharmacokinetic improvements (relative AUC/Cmax) where available.
Unfortunately, we could not obtain this information because most of the recent Clinical trials in inflammatory diseases (OA, IBD, psoriasis, COPD, sepsis) generally reported efficacy and safety outcomes but do not provide detailed pharmacokinetics like AUC and Cmax.
- Please add a dedicated safety subsection summarizing AEs, known interaction risks, and limits of current safety data.
As suggested, we have included an additional subsection in this area. (please see section 8).
Reviewer 2 Report
Comments and Suggestions for Authors
This paper compiles important information about using curcumin to treat various diseases. It is a compendium of the most significant issues related to curcumin's properties and its impact on human health. However, I had a few questions for the authors during the review process.
1) Line 67: The sentence lacks clarification as to what kind of "inflammation" is meant. Is it inflammation of an organ, the entire body, tissue, or a group of tissues?
2) Regarding lines 87 and 95–96: These sentences require an explanation of the abbreviations OA, COPD, and IBD (when they first appear in the text). It is important for the reader to know what disease entity is being referred to at this stage of the paper.
3) Lines 126–136 lack information on how the curcumin treatment was carried out in the experiments. This information is important because the results of the treatment will be discussed later. However, there is no information about how long the treatment lasted, what form the curcumin was administered in, or what the dosage was.
4) Line 178: How long did the use of curcumin have to be used before such changes occurred? No more detailed information about the course of the experiment is available.
5) Line 360: There is no information about the cell cultures involved. Similarly, line 451 lacks information about the types of cell cultures used in the studies whose conclusions are presented in subsequent paragraphs.
Author Response
This paper compiles important information about using curcumin to treat various diseases. It is a compendium of the most significant issues related to curcumin's properties and its impact on human health. However, I had a few questions for the authors during the review process.
1) Line 67: The sentence lacks clarification as to what kind of "inflammation" is meant. Is it inflammation of an organ, the entire body, tissue, or a group of tissues?
As suggested, we have included “in the body” in that sentence.
2) Regarding lines 87 and 95–96: These sentences require an explanation of the abbreviations OA, COPD, and IBD (when they first appear in the text). It is important for the reader to know what disease entity is being referred to at this stage of the paper.
As suggested, we have defined the abbreviations at the first appearance.
3) Lines 126–136 lack information on how the curcumin treatment was carried out in the experiments. This information is important because the results of the treatment will be discussed later. However, there is no information about how long the treatment lasted, what form the curcumin was administered in, or what the dosage was.
As suggested, we have included this information and also provided additional table 1.
4) Line 178: How long did the use of curcumin have to be used before such changes occurred? No more detailed information about the course of the experiment is available.
As suggested, we have included this information at this place and also in the new table 1.
5) Line 360: There is no information about the cell cultures involved. Similarly, line 451 lacks information about the types of cell cultures used in the studies whose conclusions are presented in subsequent paragraphs.
As suggested, we have included “ in the blood” for line 360, and (HT-29) at 467.
Reviewer 3 Report
Comments and Suggestions for Authors
The manuscript presents a comprehensive and timely review on the therapeutic potential of curcumin in various inflammatory diseases. The topic is highly relevant, and the extensive literature review highlights both preclinical and clinical studies, offering a strong overview for readers. Though, some of the concerns are addressed below:
- While the manuscript is extensive, much of the discussion reiterates previously known findings. there is a clear gap of novelty/scope of this review article. A stronger emphasis on recent developments (last 5 years) and critical gaps in knowledge would increase novelty.
- Authors need more critical evaluation. The review often summarizes studies without sufficient critical appraisal. A section comparing contradictory results, limitations of current evidence, and the strength/quality of clinical trials would strengthen the manuscript.
- The paper would benefit from a clearer discussion on how close curcumin is to clinical translation. For example, what specific indications appear most promising, and which require larger randomized controlled trials?
- Although the authors state PubMed was searched, the methodology (inclusion/exclusion criteria, time frames) should be described more rigorously for transparency.
- several sentences are repetitive and lengthy. Minor grammatical errors and complex sentence structures should be polished for a smoother reading experience. Some sections could be shortened or combined to avoid redundancy and maintain reader engagement.
- authors should define all abbreviations at their first mention in the tex. e.g.- in introduction authors mentioned OA, COPB, IBD etc without full forms.
- adding some summary table with clinical trial outcomes, formulations tested etc would greatly improve accessibility for readers.
- Although the safety profile of curcumin is generally favorable, the review could benefit from highlighting any reported adverse effects or toxicities, especially for long-term or high-dose usage.
- A concluding section highlighting discussion on regulatory hurdles, manufacturing challenges of nano-curcumin, and cost-benefit analyses, potential combination therapies would enhance practical relevance.
- Also, Highlight any ongoing large-scale clinical trials or needed future studies to establish efficacy conclusively.
Overall, these suggestions aim to strengthen the scientific rigor, clarity, and clinical applicability of the review article on curcumin in inflammatory diseases. With these improvements, the manuscript would make a valuable contribution to the literature.
Author Response
The manuscript presents a comprehensive and timely review on the therapeutic potential of curcumin in various inflammatory diseases. The topic is highly relevant, and the extensive literature review highlights both preclinical and clinical studies, offering a strong overview for readers. Though, some of the concerns are addressed below:
- While the manuscript is extensive, much of the discussion reiterates previously known findings. there is a clear gap of novelty/scope of this review article. A stronger emphasis on recent developments (last 5 years) and critical gaps in knowledge would increase novelty.
As suggested, most of the data discussed is limited to last 5 years to current. In case such studies are not available for particular inflammatory complication, then we extended it to 5 to 10 years.
- Authors need more critical evaluation. The review often summarizes studies without sufficient critical appraisal. A section comparing contradictory results, limitations of current evidence, and the strength/quality of clinical trials would strengthen the manuscript.
As suggested, we have included additional studies, safety prolife and adverse events in the revised manuscript. Also included three additional tables that show comprehensive information on the clinical trials.
- The paper would benefit from a clearer discussion on how close curcumin is to clinical translation. For example, what specific indications appear most promising, and which require larger randomized controlled trials?
As suggested, we have included requirement for large randomized controlled clinical trials in the conclusion sections. Current evidence suggests the strongest promise in osteoarthritis, ulcerative colitis, and sepsis, where multiple RCTs and biomarker data indicate consistent benefits. In contrast, conditions like COPD/asthma, psoriasis, and atopic dermatitis show encouraging but preliminary findings that require larger, longer RCTs to confirm efficacy and safety.
- Although the authors state PubMed was searched, the methodology (inclusion/exclusion criteria, time frames) should be described more rigorously for transparency.
Since this is a narrative review article, we have not provided all the information. However, as suggested, we have included this information in the last paragraph of the introduction.
- several sentences are repetitive and lengthy. Minor grammatical errors and complex sentence structures should be polished for a smoother reading experience. Some sections could be shortened or combined to avoid redundancy and maintain reader engagement.
As suggested, we have carefully reviewed the language and made appropriate corrections.
- authors should define all abbreviations at their first mention in the tex. e.g.- in introduction authors mentioned OA, COPB, IBD etc without full forms.
As suggested, we have defined all the abbreviations at their first use.
- adding some summary table with clinical trial outcomes, formulations tested etc would greatly improve accessibility for readers.
As suggested, we have included new table 1.
- Although the safety profile of curcumin is generally favorable, the review could benefit from highlighting any reported adverse effects or toxicities, especially for long-term or high-dose usage.
As suggested, we have discussed these points in the new subsection 8.
- A concluding section highlighting discussion on regulatory hurdles, manufacturing challenges of nano-curcumin, and cost-benefit analyses, potential combination therapies would enhance practical relevance.
As suggested, we have included this information in the conclusions. (please see highlighted section)
- Also, Highlight any ongoing large-scale clinical trials or needed future studies to establish efficacy conclusively.
As suggested, we have included this information in the conclusions. (please see highlighted section)
Round 2
Reviewer 1 Report
Comments and Suggestions for Authors
Thank you for your repaired the paper. And you have addressed my comments carefully.